# Loss of Fsr quorum sensing promotes biofilm formation and worsens outcomes in enterococcal infective endocarditis

Haris Antypas [1] ✉, Verena Schmidtchen [2], Willy Isao Staiger [2], LI Yanhong[3,4], Rachel Jing Wen Tan[1], Kenneth Kok Fei NG[1], Cheryl Jia Yi Neo[1], Shalome Meera Radhesh [1], Frederick Reinhart Tanoto [1], Ronni Anderson Gonçalves da Silva [1,5], Cristina Colomer-Winter [6], Sara Doina Schütz[2], Joachim Kloehn [6], Logeshwari Muthualagu Natarajan[1], Caroline Manzano[6], Jun Jie Wong[1], Kevin Pethe [1,5,7,8], Barbara Hasse[2], Silvio Daniel Brugger [2], Siu Ling Wong [7,9], Daria Van Tyne [3], Annelies S. Zinkernagel [2] & Kimberly A. Kline [1,5,6] ✉

Infective endocarditis (IE) is a severe heart infection caused predominantly by Gram-positive bacteria forming biofilm on heart valves. While biofilm formation is central to disease progression, the underlying bacterial mechanisms remain poorly understood. Here, we identify the Fsr quorum sensing (QS) system of *Enterococcus faecalis* as an unexpected negative regulator of biofilm and pathogenesis in IE. Using microfluidic and in vivo models, we show that blood flow prevents Fsr activation in early IE, with Fsr induction occurring only later, once bacteria form biofilm microcolonies and become shielded from flow. Deletion of Fsr promotes robust biofilm growth, driven partly through the downregulation of GelE and SprE proteases, reprograms metabolism by upregulating *lrgAB* to enhance pyruvate utilization, and increases gentamicin tolerance in vivo. Furthermore, we show that GelE cleaves the human pro-IL-1β into an active form, suggesting a species-specific mechanism for inflammation modulation by QS. In support of these findings, analysis of IE patient cohorts shows that naturally occurring Fsr-deficient *E. faecalis* strains are associated with prolonged bacteremia. Overall, our findings provide insights into how host blood flow impacts QS activation, which, in turn, regulates pathogenesis in IE, and highlight the Fsr QS as a potential determinant of clinical disease course.

Quorum sensing (QS) is an interbacterial communication system that regulates biofilm formation and other group behaviors through the production, secretion, and sensing of autoinducer signaling molecules[1]. QS is activated when autoinducers accumulate to a threshold concentration, correlated with increased bacterial population density[1], a condition commonly encountered in biofilms within natural habitats. When activated, QS can promote or inhibit gene expression involved in the extracellular matrix polysaccharide and protein synthesis, protease secretion, and eDNA release[2,3]. This regulation can drive the maturation or dispersal of biofilms[2]. However, QS is not solely regulated by population density. Mechanical and chemical cues, such as fluid flow, habitat architecture, and nutrient availability, also impact its activation[4–10]. For instance, fluid flow can disperse autoinducers through advection in vitro, thereby preventing QS

activation[4,5]. Conversely, bacteria growing in confined and structurally complex in vitro habitats can entrap autoinducers, facilitating QS activation[4,6–8]. Furthermore, despite QS being widespread across the bacterial kingdom, QS-defective mutants in bacterial species such as *Enterococcus faecalis*, *Staphylococcus epidermidis*, and *Pseudomonas aeruginosa* have been isolated from patients, indicating that QS activation may not always confer an advantage[11–14]. This genomic variation combined with the diverse mechanical and chemical cues present in the host, raises a central question: How does QS regulate biofilm formation across different host microenvironments, and how does this regulation shape infection outcomes?

During infection, bacteria encounter diverse cues from the host microenvironment, such as heterogeneous fluid flow, nutrient availability, and tissue architecture[15]. One host microenvironment where bacteria encounter all these cues is the heart valves, which direct blood flow through the heart chambers and major arteries[16]. Despite experiencing one of the highest blood flow rates in the body, heart valves are not exempt from bacterial colonization and infection. Mechanical or inflammatory lesions on the heart valves can render them susceptible to infective endocarditis (IE)[17]. These lesions trigger the accumulation of platelets and fibrin, leading to thrombus formation at the site of the valvular damage[18]. Thrombi serve as a substrate for adhesion by Gram-positive bacteria, such as *E. faecalis, S. aureus*, and *Streptococcus spp*[19–23]. Once bacteria attach to the thrombus, they can proliferate into a biofilm, leading to the infiltration of inflammatory cells and eventually to the formation of a septic vegetation[24]. The inflammatory response is often insufficient to clear the infection[25–27]. Treatment of IE typically requires a combination of intravenous antibiotic therapy for up to 6 weeks[28], but due to the high prevalence of multidrug-resistant bacteria and recalcitrance of biofilms to antibiotics, 50% of patients will also undergo surgery to remove infected tissue and replace the damaged valves[29]. Despite these treatment approaches, global IE deaths have risen from 28,754 in 1990 to 66,322 in 2019, with age-standardized mortality of IE increasing from 0.73 to 0.87 per 100,000 people over the same period[30]. This alarming increase highlights the need to better understand how bacteria adapt to the vegetation microenvironment to improve the diagnosis and treatment of IE.

Limited insights into the biofilm-host interplay on cardiac vegetations constrain advances in the diagnosis and treatment of IE. Most studies have focused on initial bacterial adhesion to the thrombus exterior, a microenvironment that differs significantly from the conditions encountered within the vegetation during biofilm maturation. The vegetation exterior is exposed to intense shear stress forces due to blood flow, while the interior is shielded from them. This differential exposure to blood flow may influence QS activity and nutrient availability, depending on bacterial location at the different stages of infection. In this study, we investigated the hypothesis that the heterogeneous microenvironment of IE vegetations critically influences biofilm development and infection outcome in IE caused by *E. faecalis*, the third most common causative agent of this infection[31,32]. While the virulence factors that facilitate initial adhesion to thrombus in enterococcal IE have been identified[33–40], the regulation of biofilm formation and its impact on disease progression remain largely overlooked. Here, using microfluidic and in vivo models, we show that fluid flow prevents Fsr activation, revealing a spatiotemporal regulation of QS as vegetations grow. We identify Fsr as a negative regulator of biofilm growth in vivo, exerting its control by upregulating *gelE* and *sprE* and by limiting *lrgAB* expression. We also demonstrate that loss of the Fsr QS system, commonly observed in IE clinical isolates[12,13], increases tolerance to gentamicin treatment in vivo and correlates with prolonged bacteremia and high disease severity in individuals with enterococcal IE. Collectively, these findings provide new insights into the role of QS in IE biofilm growth, antibiotic tolerance, and infection outcomes.

## Results

### Fluid flow and microcolony size regulate Fsr QS activation in IE

Vegetations in IE present a heterogeneous microenvironment, where the exterior is exposed to high shear stress and advection from blood flow, while the interior remains shielded. At the onset of IE, bacteria in the bloodstream attach to the vegetation exterior and become directly exposed to blood flow. We therefore hypothesized that blood flow might trigger physiological adaptations in *E. faecalis* that favor the establishment of IE. To identify these adaptations, we exposed surface-attached *E. faecalis* OG1RF to a pulsatile flow of media for 30 minutes in a microfluidic system. This flow generated a shear stress of 20 dynes/cm$^2$, simulating the shear stress on aortic valve leaflets[41–43]. Compared to surface-attached bacteria that were incubated without flow, transcriptomic analysis revealed 59 genes upregulated and 167 genes downregulated under flow ($\log_2$FC > 1.0, FDR < 0.05) (Fig. 1A, Supplementary Data 1). Gene ontology enrichment analysis revealed an overrepresentation of genes linked to ATPase activity, glycolytic processes, and oxidative stress (Fig. 1B), suggesting that *E. faecalis* adapts its metabolism to meet energy demands and adapt to different oxygen levels under flow. Notably, transcription of the *fsr* locus (*fsrABDC*) and its associated regulon (*gelE, sprE, entV, RSO4585*)[38,44,45] were significantly decreased in the presence of fluid flow (Fig. 1A). The *fsr* locus encodes the Fsr quorum sensing (QS) system, which is activated when the extracellular concentration of the autoinducer peptide (AIP) gelatinase biosynthesis-activating pheromone (GBAP) exceeds a critical concentration and is therefore typically activated in direct proportion with bacterial population density[46]. Given the role of GBAP concentration in Fsr activation, we investigated whether *fsr* downregulation under fluid flow was attributed to GBAP advection. To distinguish between advection- and shear stress-mediated transcriptional regulation, we exposed surface-attached bacteria to low (1 dynes/cm$^2$), intermediate (10 dynes/cm$^2$), and high shear stress (20 dynes/cm$^2$). Quantitative PCR showed that transcription of *fsr*-associated genes was decreased 3- to 4-fold after 30 min, regardless of the shear stress magnitude applied (Fig. 1C), demonstrating that advection rather than mechanical forces regulates their expression. Taken together these data show that fluid flow downregulates quorum sensing gene expression and triggers physiological adaptations in *E. faecalis*.

To directly demonstrate the effect of flow on QS activation in vivo, we constructed a Dasher GFP reporter plasmid under the control of the *gelE* promoter (P*gelE*-GFP) (Figure S1A-H), which is transcriptionally upregulated by FsrA[47]. Using a rat model of IE, we infected animals with OG1RF WT transformed with P*gelE*-GFP (WT P*gelE*-GFP) or with the constitutive promoter control plasmid P*cfb*-GFP (WT P*cfb*-GFP)[48], followed by an IV injection of spectinomycin to select for plasmid maintenance in vivo (Figure S1I, J). Microcolonies in aortic valve vegetations harvested at 24 h post-infection (hpi) were examined by microscopy, and GFP expression, size, and distance from the vegetation surface were quantified. Surface microcolonies exhibited lower gelatinase promoter expression compared to embedded microcolonies (Fig. 1D, E, *p* = 0.0239), and small microcolonies exhibited lower gelatinase promoter expression compared to large microcolonies (Fig. 1F, *p* = 0.001). In WT P*cfb*-GFP infection, only small aggregates were detected and CFU counts in vegetations were reduced compared to WT P*gelE*-GFP infection (Figure S1I, J), suggesting that constitutive GFP overexpression impaired strain fitness in the host. Nevertheless, these small aggregates fluoresced brightly regardless of their location (Figure S1K), confirming the specificity of the P*gelE*-GFP reporter. Overall, these findings show that exposure to fluid flow and microcolony size regulate QS activation in IE.

### Early colonization of vegetation is independent of the Fsr QS system

Fluid flow restricts Fsr activation in superficial microcolonies, suggesting that the QS system may be dispensable for *E. faecalis* to

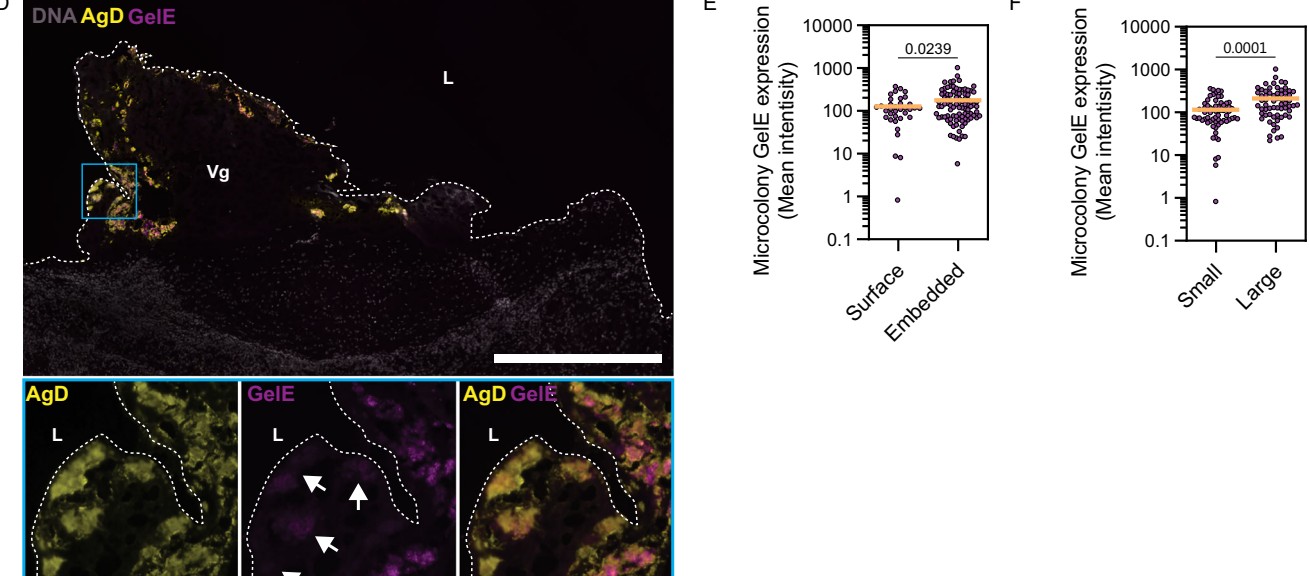

**Fig. 1 | Fluid flow and microcolony size regulate Fsr QS activation in IE. A** Volcano plot showing differentially expressed genes in *E. faecalis* OG1RF after exposure to fluid flow for 30 min compared to static conditions. Genes exhibiting log$_2$FC > 1.0 with FDR < 0.05 are shown in black. *fsr* locus genes and associated regulon are labeled. N = 3. **B** Gene Ontology (GO) enrichment analysis showing the most significantly enriched biological processes (p < 0.05) in *E. faecalis* when subjected to fluid flow. Statistical significance was assessed with the Wallenius Method. **C** Mean relative gene expression under different magnitudes of shear stress compared to static conditions assessed by qPCR. N = 5, error = SEM. Statistical significance was determined using a one-way ANOVA test and Tukey's multiple comparison test on ΔCt values. **D** Vegetation (Vg) section infected with WT P*gelE*-GFP and harvested at 24 hpi. The blue inset magnifies a vegetation area where Fsr quorum sensing activation is reduced in microcolonies located on its surface

(arrows). Image was acquired with epifluorescence tiling microscopy and stained for DNA and *Enterococcus*-specific Group D *Streptococcus* antigen (AgD). L = lumen, dashed line = vegetation boundary. Representative image is shown from *n* = 3 animals from N = 1. Scale = 500 μm. **E, F** Quantification of gelatinase (GelE) expression in microcolonies from vegetations infected with WT P*gelE*-GFP and harvested at 24 hpi. A total of 129 microcolonies from 2–3 vegetation sections per rat (*n* = 3) were analyzed for gelatinase mean intensity, distance from vegetation surface, and area. Mean is shown. Statistical significance between 35 surface and 94 embedded microcolonies, and between 65 small and 64 large microcolonies was determined with a two-tailed unpaired t-test with Welch's correction; *n* = animals per group, N = independent experiments, *ns* not significant (*p* ≥ 0.05). Exact *p* values are reported in the figure. Source data are provided as a Source Data file.

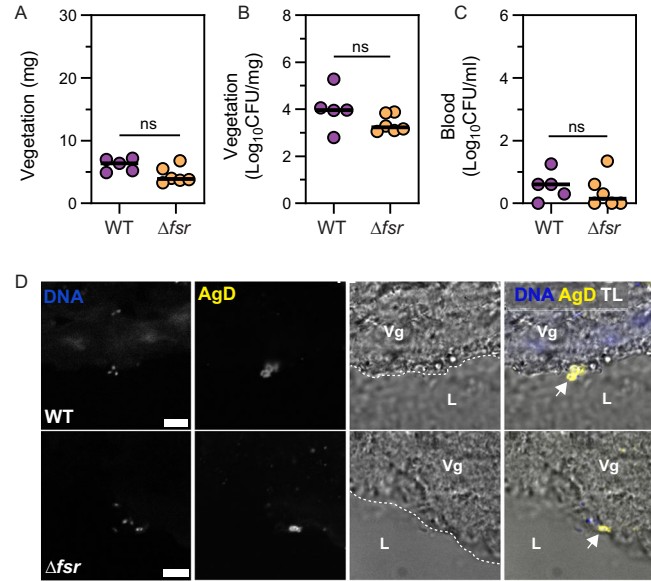

**Fig. 2 | Early colonization of vegetation is independent of the Fsr QS system.**
**A–C** Vegetation weight (**A**), vegetation CFU (**B**), and blood CFU (**C**) at 6 hpi. Median shown for $n = 5$ (WT) and $n = 6$ ($\Delta fsr$), $N = 1$. Statistical significance was determined using a two-tailed Mann-Whitney test. **D** *E. faecalis* adhesion on vegetation (Vg) surface at 6 hpi captured with laser scanning confocal microscopy (LSCM), stained for DNA and *Enterococcus*-specific Group D *Streptococcus* antigen (AgD) and merged with transmitted light (TL) images. *L* lumen, dashed line = vegetation boundary, arrow = bacteria. Representative images are shown from $n = 2$, $N = 2$. Scale = 5 μm; $n$ = animals per group, $N$ = independent experiments, *ns* not significant ($p \geq 0.05$). Source data are provided as a Source Data file.

establish IE. To test this, we harvested vegetations at 6 hpi infected with either the OG1RF parental wild type or $\Delta fsrABDC$ strain (hereafter called WT and $\Delta fsr$ respectively). We found no statistically significant differences in vegetation weight or bacterial colony forming units (CFU) in vegetations or blood between these two strains (Fig. 2A–C). Imaging of vegetations at 6 hpi revealed the presence of single superficial bacteria for both WT and $\Delta fsr$ strains (Fig. 2D). These results agree with earlier findings showing no difference in the ability of OG1RF and a *fsrB* deletion mutant to induce IE in a rat model[49], and they are consistent with our in vivo P*gelE*-GFP reporter findings, further supporting that Fsr QS is not activated in superficial and sparsely colonized vegetations of early infection.

## Absence of Fsr QS system promotes biofilm growth in late IE
As IE progresses, vegetations enlarge through bacterial replication and deposition of platelets and fibrin. *E. faecalis* microcolonies embedded in the interior of mature vegetations become shielded from blood flow, favoring Fsr QS activation. Consistent with this, *fsrC* expression was higher compared to the bacterial inoculum in WT-infected vegetations harvested from rats at 24 and 72 hpi (Fig. 3A, $p = 0.0002$ and $p = 0.0074$ respectively). To investigate how QS activation affects virulence at later stages of infection, we infected rats with the WT and $\Delta fsr$ strains. We found that 72-h vegetations infected with $\Delta fsr$ were larger compared to WT (Fig. 3B, $p = 0.0167$). Moreover, $\Delta fsr$-infected animals exhibited a higher bacterial load within each vegetation, compared to WT-infected animals (Fig. 3C, $p = 0.0002$). Immunofluorescence on vegetation sections corroborated this finding (Fig. 3F), showing that $\Delta fsr$ biofilms covered a larger vegetation area compared to WT (Fig. 3D, $p = 0.0026$), ranging between 21–42 % for $\Delta fsr$ compared to 5–15 % for WT. Furthermore, individual $\Delta fsr$ microcolonies within vegetations exhibited a larger cross-sectional area compared to WT (Fig. 3E and G, $p < 0.0001$). Growth curves comparing WT and $\Delta fsr$

strains showed no growth advantage for the mutant strain in serum-supplemented BHI (Figure S2A). Gelatinase has been shown to degrade polymerized fibrin[25,50,51], a main component of vegetations[25,51]. This degradation has been proposed as a mechanism for bacteria to disperse from the vegetations into the bloodstream[50]. However, we did not observe differences for $\Delta fsr$ dissemination to the blood, liver, and spleen (Figure S2B–D), suggesting that impaired dispersal is unlikely to explain the increased vegetation size and bacterial burden observed in the absence of Fsr. Overall, these data demonstrate that Fsr plays a role in limiting biofilm and vegetation size in late IE.

## Absence of *gelE* and *sprE* contributes to biofilm growth in late IE
The *fsr* regulon includes *gelE* and *sprE*, which encode the metalloprotease gelatinase and serine protease respectively. These secreted proteases are major virulence factors in *E. faecalis*, affecting cell division, autolysis, adhesion, opsonization, and extracellular matrix composition[44,50,52–54]. Thus, we hypothesized that Fsr might mediate vegetation and bacterial growth control via gelatinase and/or serine protease activity. As expected for Fsr-regulated genes, similar to *fsrC* expression (Fig. 3A), *gelE* and *sprE* expression was higher at 72 hpi in WT-infected vegetations compared to the inoculum (Fig. 4A, B, $p = 0.018$ and $p = 0.0055$ respectively). When we infected rats with a $\Delta gelE$ mutant, we found no difference in vegetation weight compared to WT infection (Fig. 4C). However, we observed an increase in vegetation CFU for $\Delta gelE$ (Fig. 4D, $p = 0.0155$, Median$_{\Delta gelE}$ – Median$_{WT}$ = 0.290), albeit less pronounced compared to $\Delta fsr$ (Median$_{\Delta fsr}$ – Median$_{WT}$ = 0.468). Growth curves comparing WT and $\Delta gelE$ strains showed no growth advantage for the mutant strain in vitro (Figure S2E). Despite a previously suggested role of gelatinase in promoting biofilm dispersion[50], we observed no differences in systemic spread of bacteria between WT and $\Delta gelE$ (Figure S2F-H). Next, we investigated the spatial organization of the increased bacterial load in $\Delta gelE$ vegetations. Immunofluorescence showed increased biofilm coverage in $\Delta gelE$-infected vegetation sections, ranging between 11–38 % of the total vegetation area, whereas WT biofilm coverage ranged between 6–12 % (Fig. 4E, G). Additionally, individual microcolonies in $\Delta gelE$ vegetations were larger compared to WT (Fig. 4F, H, $p < 0.0001$). Similar to our observation for $\Delta gelE$ infection, $\Delta sprE$-infected vegetations exhibited a comparable weight to WT (Fig. 4I) but an increase in CFU (Fig. 4J, $p = 0.0317$, Median$_{\Delta sprE}$ – Median$_{WT}$ = 0.324). Growth curves comparing WT and $\Delta sprE$ strains showed no growth advantage for the mutant strain during exponential phase in vitro (Figure S2I). We observed an increase in spleen CFU in $\Delta sprE$-infected animals but no difference in blood and liver CFU compared to WT (Figure S2J–L). Infection with the double deletion mutant $\Delta gelE\Delta sprE$ exhibited a similar vegetation weight (Fig. 4K) and an increased vegetation CFU compared to WT (Fig. 4L, $p = 0.0229$, Median$_{\Delta gelE\Delta sprE}$ – Median$_{WT}$ = 0.376). Despite the lack of both proteases, the increase was not as pronounced as in the absence of *fsr*. Growth curve comparison between WT and $\Delta gelE\Delta sprE$ showed no difference in growth during exponential phase (Figure S2M). Blood, spleen, and liver CFU were significantly lower compared to WT (Figure S2N–P), indicating a defect in systemic spread. However, this dispersion defect was absent in $\Delta fsr$ infection (Figure S2B–D), suggesting that other factors regulated by Fsr might compensate for impaired dispersal when both *gelE* and *sprE* are downregulated. Collectively, our data show that gelatinase and serine protease contribute partially to biofilm growth control conferred by the *fsr* locus in IE vegetations and suggest that additional *fsr* associated factors likely also contribute.

## Fsr absence enhances virulence in IE via upregulation of *lrgAB*-mediated pyruvate utilization
Since enhanced biofilm growth in the absence of the *fsr* locus was only partly explained by the absence of gelatinase and serine protease, we hypothesized that the absence of *fsr* might also enhance biofilm

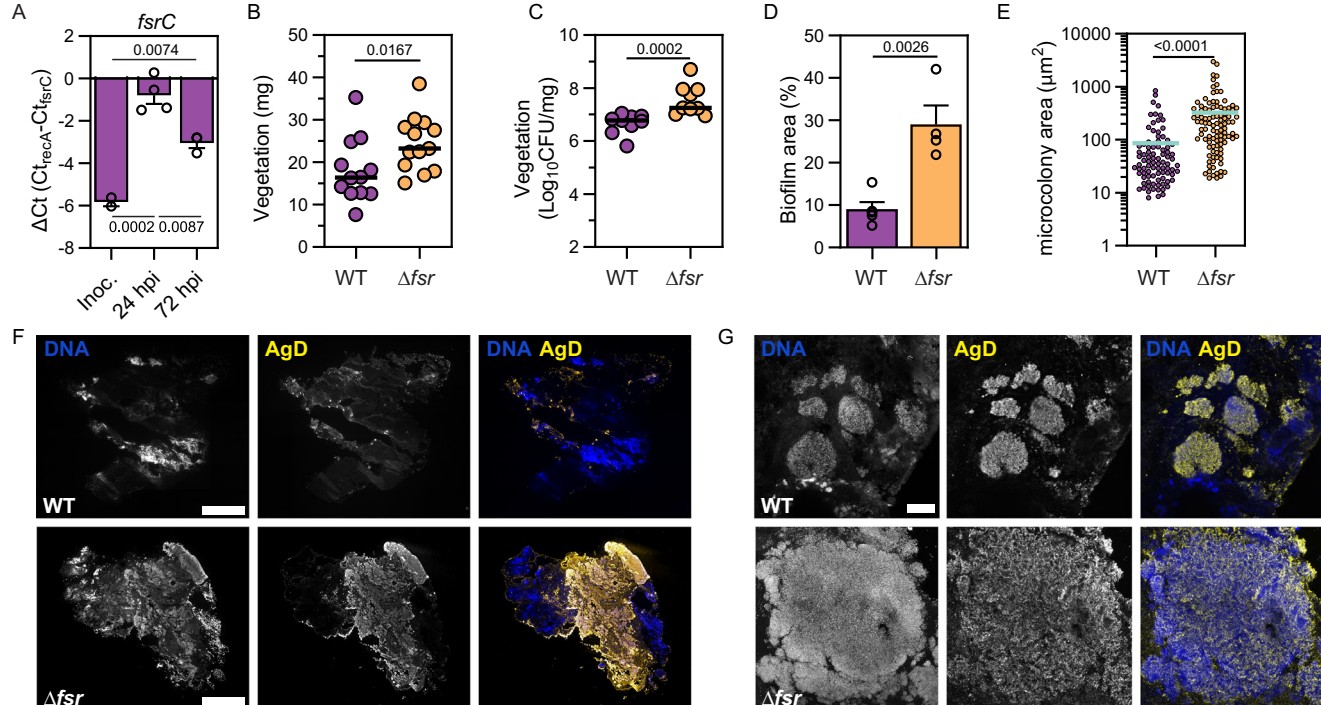

**Fig. 3 | Absence of Fsr QS system promotes biofilm growth in late IE. A** qPCR analysis showing mean *fsrC* expression of the WT strain in the inoculum and vegetations at 24 and 72 hpi. *fsrC* expression is presented as ΔCt values normalized to the expression of the reference gene *recA*. Vegetation cDNA samples derived from n = 4 (24 hpi) and *n* = 3 (72 hpi), N = 2. Error = SEM. Statistical significance was determined using a one-way ANOVA test and Tukey's multiple comparison test. **B** Vegetation weight at 72 hpi. Median shown for *n* = 12 (WT) and *n* = 13 (Δ*fsr*), N = 3. Statistical significance was determined using a two-tailed Mann-Whitney test. **C** Vegetation CFU at 72 hpi. Median of *n* = 9 animals per group from N = 2 is shown. Statistical significance was determined using a two-tailed Mann-Whitney test. **D** Mean biofilm area (%) of vegetation sections shown for *n* = 5 (WT) and *n* = 4 (Δ*fsr*), N = 2. Biofilm area was determined by calculating the antigen D (AgD) positive area relative

to total vegetation area. Error = SEM. Statistical significance was determined using a two-tailed t-test. **E** Distribution and median of intravegetational microcolony area from *n* = 3 (WT) and *n* = 2 (Δ*fsr*), N = 2. A total of 88 microcolonies for WT and 101 microcolonies for Δ*fsrABDC* were quantified. Statistical significance was determined using a two-tailed Mann-Whitney test. **F** Tile images of whole vegetation sections from 72 hpi captured with fluorescence microscopy, stained for DNA and AgD. Representative sections shown from *n* = 5 (WT) and *n* = 4 (Δ*fsr*), N = 2. Scale = 500 μm. **G** Z-projections of biofilm microcolonies from 72 hpi captured with LSCM stained for DNA and AgD. Representative images shown from *n* = 5 (WT) and n = 4 (Δ*fsr*), N = 2. Scale = 20 μm; *n* = animals per group, N = independent experiments. Exact *p* values are reported in the figure. Source data are provided as a Source Data file.

growth by optimizing its ability to exploit available nutrients and adapt to the microenvironment of the vegetation. We therefore compared the *E. faecalis* transcriptome of 72 hpi vegetations infected with WT and Δ*fsr* strains. We found 292 genes upregulated and 83 genes downregulated (Log2FC ≥ 1, FDR < 0.05) in Δ*fsr* compared to the WT strain (Fig. 5A, and Supplementary Data 2). As expected, the Fsr regulon comprising *gelE* (Log2FC = −8.2), *sprE* (Log2FC = −8.1), *entV* (Log2FC = −7.7), and *OG1RF_RS04585* (Log2FC = −8.8) exhibited the most pronounced downregulation. Among the upregulated genes, antiholin-like murein hydrolase modulator *lrgA* (OG1RF_RS12570) and antiholin-like protein *lrgB* (OG1RF_RS12565) exhibited the greatest increase in expression (Log2FC = 5.0) (Fig. 5A). To identify other biological processes that may be significantly enriched in the absence of the *fsr* locus in vivo, we conducted a gene ontology (GO) enrichment analysis. We found that differentially expressed genes were significantly enriched in processes related to nucleotide biosynthesis, sugar transport and carbohydrate metabolism (Fig. 5B). One of the most enriched categories was the phosphoenolpyruvate-dependent sugar phosphotransferase systems (PTS), which are involved in the uptake and phosphorylation of specific carbohydrates from the extracellular environment. Thus, we concluded that the Fsr QS system likely represses multiple metabolic pathways in vegetations, and its absence leads to the derepression and enhanced utilization of various substrates available in the microenvironment of the vegetation.

*lrgAB* homologs in *Bacillus subtilis*, *S. aureus*, and *Streptococcus mutans* are involved in transporting pyruvate[55–58], which is present in

blood[59]. To investigate whether *lrgAB* plays a similar role in *E. faecalis*, we cultured OG1RF WT, Δ*fsr*, *lrgA*::Tn, and *lrgB*::Tn strains in minimal medium M1, with and without pyruvate, under both aerobic and anaerobic conditions, to account for potentially different oxygen availability within the large 72-h vegetations. Despite pyruvate supplementation, growth of *lrgA*::Tn was impaired compared to WT under both conditions (Fig. 5C, D), with the impairment being more pronounced under anaerobic conditions, whereas the *lrgB*::Tn strain exhibited impairment only under anaerobic conditions (Fig. 5D). These growth defects were specific to pyruvate supplementation, as *lrgA*::Tn and *lrgB*::Tn reached the same growth levels to WT with glucose supplementation (Fig. 5C, D). To investigate whether the growth defects observed are linked to impaired pyruvate transport, we quantified extracellular and intracellular pyruvate in bacteria grown in M1 supplemented with pyruvate using gas chromatography-mass spectrometry (GC-MS). Under aerobic conditions, extracellular pyruvate abundance was higher in the *lrgA*::Tn strain relative to WT and *lrgB*::Tn (Fig. 5E, *p* < 0.0001), indicating that *lrgA* is involved in pyruvate import. Similarly, intracellular pyruvate abundance was higher in the *lrgA*::Tn strain relative to WT and *lrgB*::Tn (Fig. 5E, *p* = 0.0051 & *p* = 0.0104 respectively), indicating that *lrgA* may also be involved in the downstream catabolism of pyruvate. Under anaerobic conditions, extracellular pyruvate abundance was comparable across all strains (Fig. 5F), whereas intracellular pyruvate abundance was significantly higher in *lrgB*::Tn compared to WT and *lrgA*::Tn (Fig. 5F, *p* = 0.0145 & *p* = 0.0028), indicating that *lrgB* might play a role in pyruvate

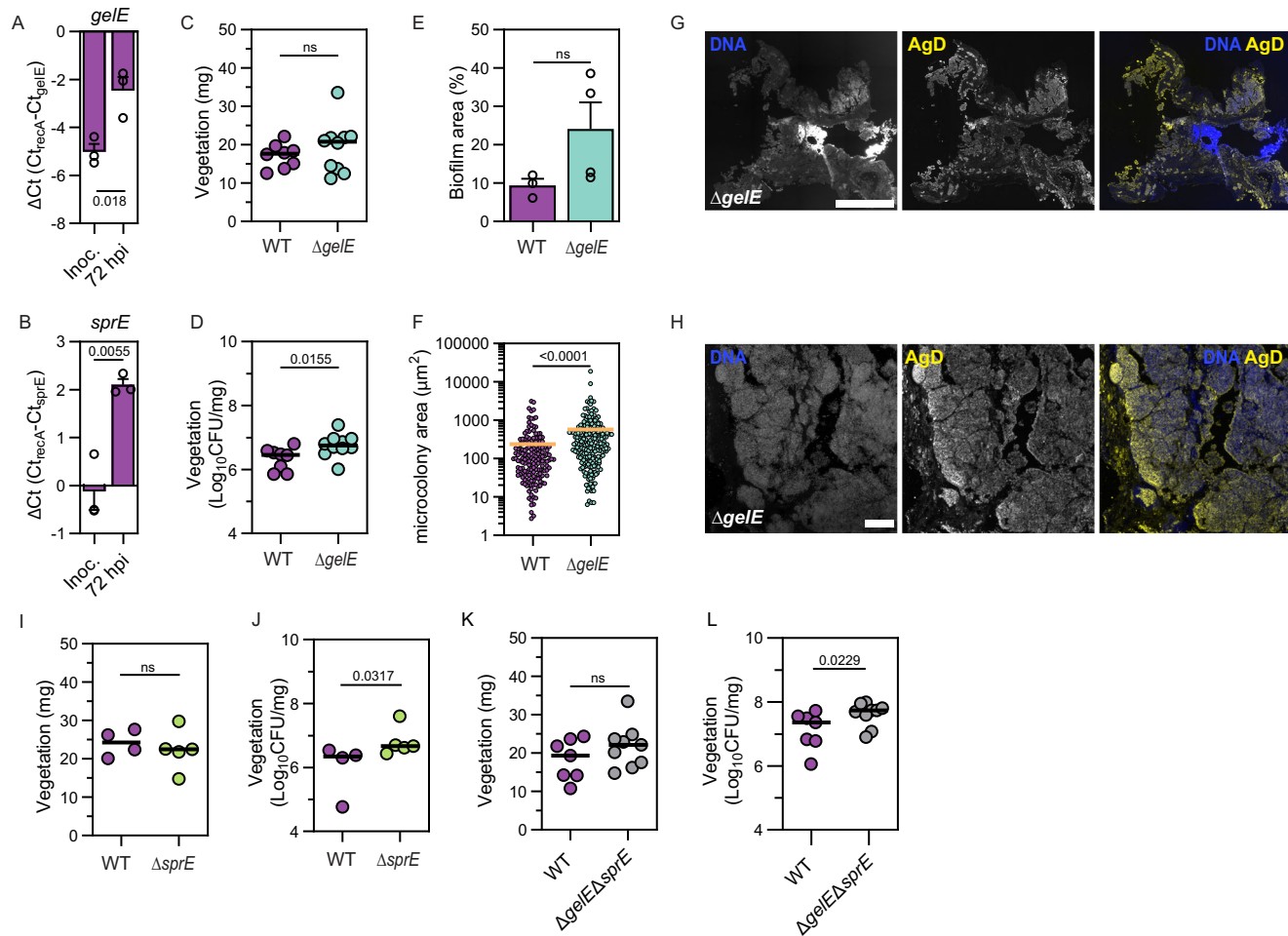

**Fig. 4 | Absence of *gelE* and *sprE* contributes to biofilm growth in late IE.**
**A**, **B** qPCR analysis showing mean *gelE* and *sprE* expression of the WT strain in the inoculum and vegetations at 72 hpi. *gelE* expression is presented as ΔCt values normalized to the expression of *recA*. Vegetation cDNA samples derived from n = 3, N = 2. Error = SEM. Statistical significance was determined using a two-tailed sample t-test. **C**, **D** Vegetation weight (**C**) and CFU (**D**) at 72 hpi. Median shown for *n* = 8 (WT) and *n* = 10 (Δ*gelE*), N = 2. Statistical significance assessed with a two-tailed Mann-Whitney test. **E** Mean biofilm area (%) per vegetation section shown from *n* = 4, N = 2. Biofilm area was determined by calculating the antigen D positive area to total vegetation area. Error = SEM. Statistical significance was assessed with t-test. **F** Distribution and median of intravegetational microcolony area shown from *n* = 3 (WT) and *n* = 4 (Δ*gelE*), N = 2. A total of 168 microcolonies for WT and 221 microcolonies for Δ*gelE* were quantified. Statistical significance was determined using a two-tailed Mann-Whitney test. **G** Epifluorescence microscopy tile images of whole vegetation sections from 72 hpi, stained for DNA and antigen D (AgD). Representative section shown from *n* = 4, N = 2. Scale = 500 μm. **H** Z-projection of biofilm microcolonies from 72 hpi captured with LSCM stained for DNA and AgD. Representative image shown from n = 4, N = 2. Scale = 20 μm. **I–L** Vegetation weight and CFU at 72 hpi. Median of *n* = 4 (WT) and *n* = 5 (Δ*sprE*), N = 1 for (**I**) and (**J**), and *n* = 7 (WT) and *n* = 9 (Δ*gelE*Δ*sprE*), N = 2 for (**K**) and (**L**) is shown, with statistical significance assessed with a two-tailed Mann-Whitney test; *ns* not significant (*p* ≥ 0.05), *n* = animals per group, N = independent experiments. Exact p-values are reported in the figure. Source data are provided as a Source Data file.

catabolism specifically under oxygen-limiting conditions. As pyruvate lies at the intersection of multiple metabolic pathways, we next quantified extracellular and intracellular levels of lactate and alanine to determine whether *lrgAB* regulates pyruvate conversion into these metabolites. No significant differences were observed under either aerobic or anaerobic conditions (Figure S3A, B), indicating that the conversion of pyruvate to lactate and alanine remained unaffected. *lrgAB* can also restrict phage-mediated extracellular lysis in *E. faecalis* and can inhibit murein hydrolase activity in *S. aureus*, limiting autolysis and conferring tolerance to penicillins[60,61]. Our data, however, did not reveal any differences in growth between WT, *lrgA*::Tn, and *lrgB*::Tn when exposed to triton X-100 or ampicillin (Figure S3C, Table S1). We also did not observe any differences in biofilm formation in vitro between WT, *lrgA*::Tn, and *lrgB*::Tn (Figure S3D, E). Taken together, these findings support a role for *lrgAB* in pyruvate catabolism in *E. faecalis*.

Given that *lrgAB* was significantly upregulated in the absence of *fsr* during IE, we next examined its role in pathogenesis by infecting rats

with a double mutant lacking both the *fsr* and *lrg* operons (Δ*fsr*Δ*lrgAB*). Vegetation weight and CFU showed no difference compared to the WT strain (Fig. 5G, H), showing that *lrgAB* contributes to the increase in vegetation mass and bacterial load observed in the absence of QS. Spleen, liver, and blood CFU showed no difference, and growth curve of Δ*fsr*Δ*lrgAB* showed no growth defect compared to WT (Figure S3F–I). Taken together, these findings suggest that upregulation of *lrgAB* in the absence of Fsr QS might enhance pyruvate utilization and act as a positive regulator of IE pathogenesis.

### *E. faecalis* gelatinase cleaves and activates human IL-1β

Gelatinase is a secreted metalloprotease with broad substrate specificity[62–65]. Thus, we investigated whether its protease activity is modulating host factors involved in the immune response in IE. Specifically, we investigated whether gelatinase modulates inflammation by activating IL-1β, as shown previously for *Streptococcus pyogenes* SpeB and *Pseudomonas aeruginosa* LasB proteases[66,67]. We detected IL-1β in both WT and Δ*gelE*-infected vegetations at 72 hpi using

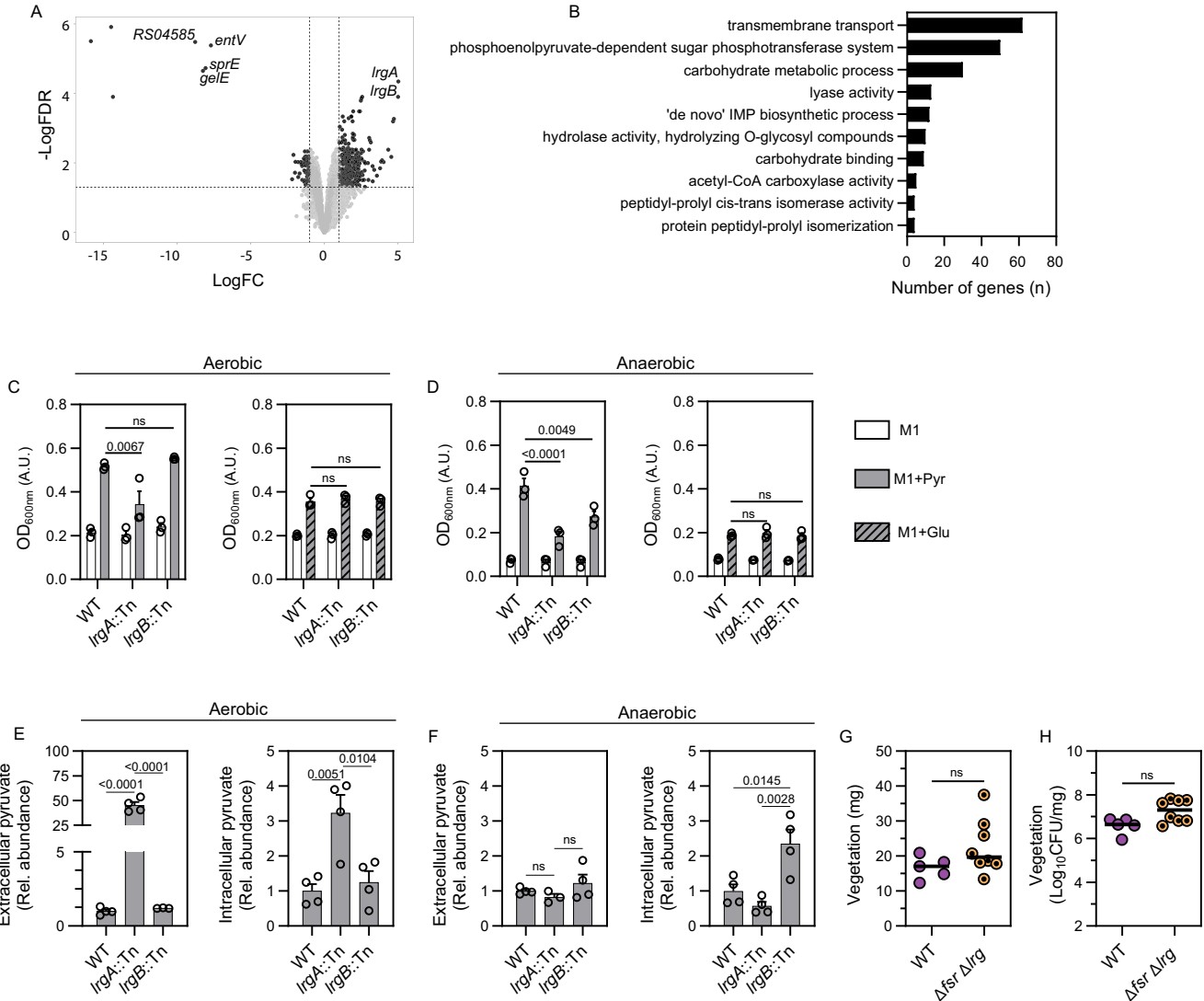

**Fig. 5 | Fsr absence enhances virulence in IE via upregulation of *lrgAB*-mediated pyruvate utilization. A** Volcano plot showing differentially expressed genes of *E. faecalis* in Δ*fsr*-infected vegetations compared to WT at 72hpi. Genes exhibiting Log₂FC ≥ 1.0 with FDR < 0.05 are shown in black. *fsr*-associated regulon and *lrgAB* are labeled. *n* = 4, N = 1. **B** Gene Ontology (GO) enrichment analysis showing the most significantly enriched biological processes in *E. faecalis* in Δ*fsr*-infected vegetations at 72 hpi. Statistical significance was assessed with the Wallenius Method. **C, D** Endpoint absorbance of OG1RF WT, *lrgA*::Tn, and *lrgB*::Tn grown in M1 media, with or without pyruvate (110 mM) or glucose (110 mM) supplementation, under aerobic (**C**) and anaerobic (**D**) conditions. Mean and SEM is shown for N = 3, two-way ANOVA was applied. **E, F** Relative abundance of extracellular and

intracellular pyruvate was measured in *E. faecalis* WT, *lrgA*::Tn, and *lrgB*::Tn cultures grown in M1 supplemented with 10 mM pyruvate under aerobic (**E**) and anaerobic (**F**) conditions using GC-MS. Mean ± SEM shown for N = 4, except extracellular pyruvate in aerobic *lrgB*::Tn and anaerobic *lrgA*::Tn cultures (N = 3). Statistical analysis was performed using one-way ANOVA followed by Tukey's multiple comparison test. **G, H** Vegetation weight (**G**) and CFU (**H**) at 72 hpi. Median shown for *n* = 5 (WT) and n = 8 (Δ*fsrΔlrg*), N = 1, with statistical significance assessed with a two-tailed Mann-Whitney test; ns = not significant (*p* ≥ 0.05), *n* = animals per group, N = independent experiments, *A.U.* arbitrary units. Exact *p* values are reported in the figure. Source data are provided as a Source Data file.

immunostaining (Fig. 6A, and Figure S10A). IL-1β fluorescence was notably stronger at the interface between the biofilm and neutrophils (blue inset in Fig. 6A) undergoing NETosis (Figure S4A), and within incoming neutrophils (orange inset in Fig. 6A & Figure S4B). Western blots using the same antibody confirmed the predominance of the IL-1β pro-form (pro-IL-1β) in WT and Δ*gelE* vegetations at 72 hpi (Fig. 6B). Given the abundance of IL-1β in direct contact with IE biofilms, we investigated whether *E. faecalis* gelatinase is involved in the activation or degradation of IL-1β. Incubation of rat pro-IL-1β with WT, Δ*gelE*, and Δ*gelE*::*gelE*^E352A, which expresses a proteolytically inactive gelatinase, resulted predominantly in gelatinase-dependent degradation of the pro-form and to a lesser extent in cleaving a fragment below 15 kDa (Fig. 6C). However, this fragment did not correspond to the canonical mature IL-1β generated by caspase-1 (17 kDa) and lacked detectable

bioactivity when we exposed the cell-free supernatants to an IL-1R reporter cell line that specifically recognizes cleaved and active IL-1β (Figure S4C). Despite the degradation of pro-IL-1β by gelatinase, quantification of immune cells using flow cytometry within vegetations showed no significant difference for either total leukocytes or the neutrophil subpopulation in Δ*gelE*-infected vegetations compared to WT at 72 hpi (Figure S4D–F).

Interestingly, incubation of human pro-IL-1β with WT, Δ*fsr*, and Δ*gelE* strains revealed gelatinase-dependent cleavage of pro-IL-1β into ~17 kDa fragments, resembling mature IL-1β following cleavage and activation by caspase-1[68] (Fig. 6D, and Figure S4G). A *sprE*-defective strain did not abolish cleavage of pro-IL-1β (Fig. 6D). Following incubation of pro-IL-1β for 6 h with WT or Δ*gelE*::*gelE*^E352A, we tested the supernatants with the IL-1R reporter cell line. Only the WT supernatant

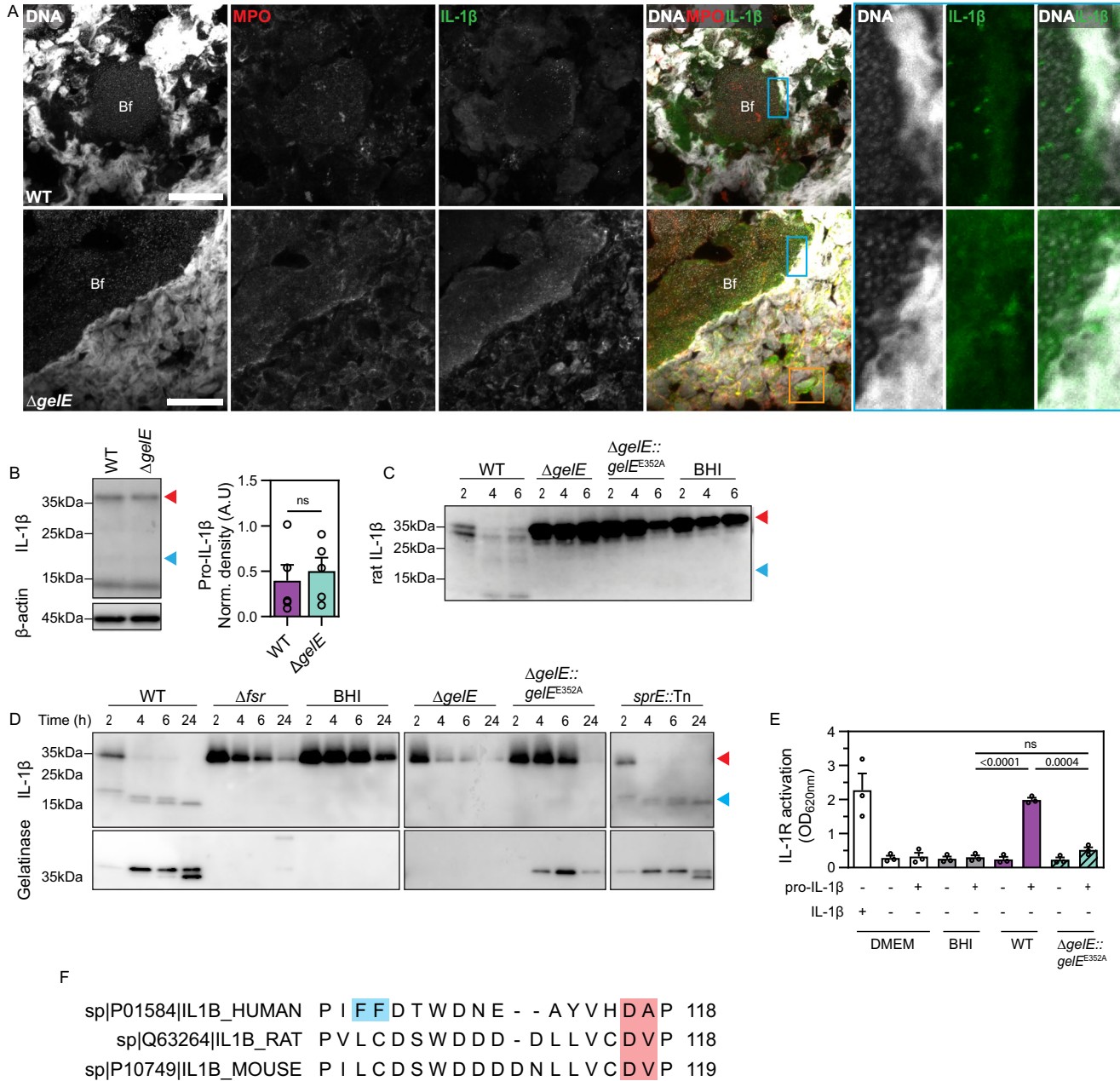

**Fig. 6 | *E. faecalis* gelatinase cleaves and activates human IL-1β. A** Z-projections of WT and ΔgelE vegetations at 72 hpi, captured with LSCM and stained for DNA, myeloperoxidase (MPO), and IL-1β. Cyan insets are highlighting the presence of IL-1β between the NETs-biofilm interface. Orange inset is shown in Fig. S4B, highlighting the colocalization of IL-1β with neutrophils. Representative images shown from *n* = 3, N = 1. Bf = biofilm, scale = 20 μm. **B** Detection and quantification of IL-1β in WT and ΔgelE vegetations at 72 hpi with western blotting. Loading control = β-actin. Mean ± SEM, n = 5 from N = 2, A.U. = arbitrary units. Statistical significance was assessed with a two-tailed t-test. **C** Rat pro-IL-1β and its cleaved fragments in supernatants from OG1RF WT and mutant strains detected by western blotting. Rat pro-IL-1β was incubated in BHI with indicated OG1RF strains for 2, 4, 6 h. BHI = Negative control with only media and pro-IL 1β. Representative blot shown from N = 2. **D** Human pro-IL-1β and its cleaved fragments in supernatants from OG1RF WT and mutant strains detected by western blotting. Human pro-IL-1β was incubated in BHI with indicated OG1RF strains for 2, 4, 6, and 24 h. Gelatinase presence was also determined in these supernatants. ΔgelE::gelE^E352A expresses proteolytically inactive gelatinase. Representative blot shown from N = 2. BHI = Negative control with only media and pro-IL 1β. **E** Mean activation of HEK-Blue IL-1R reporter cells by supernatants harvested from OG1RF WT and ΔgelE::gelE^E352A cultures with or without human pro-IL-1β at 18 h. Stimulation of cells with mature human IL-1β was used as a positive control. N = 3, Error = SEM. Statistical significance was determined with one-way ANOVA. **F** Schematic representation of gelatinase (blue) and caspase-1 cleavage sites (red) across human, rat, and mouse pro-IL-1β; *n* = animals per group, N = independent experiments, *ns* not significant (p ≥ 0.05), Red arrowhead = pro-IL-1β, blue arrowhead = mature IL-1β. Exact *p* values are reported in the figure. Source data are provided as a Source Data file.

activated the reporter cells (Fig. 6E) in a concentration-dependent manner (Figure S4H), confirming bioactive IL-1β production. To investigate the cleavage site, we excised the band corresponding to pro-IL-1β from the 0 h incubation, as well as the ~17 kDa bands observed at 6 h and 18 h and subjected them to mass spectrometry. Peptides uniquely detected at 6 h and 18 h compared to 0 h, and

enriched above a threshold abundance, were compiled (Supplementary Data 4). Among these, a peptide beginning at residue 105 was one of the most abundant, corresponding to cleavage between F104 and F105 (Supplementary Data 4). Notably, cleavage at this position is consistent with the size of the ~17 kDa fragment observed by western blot, the gelatinase substrate specificity[63], and the analogous to

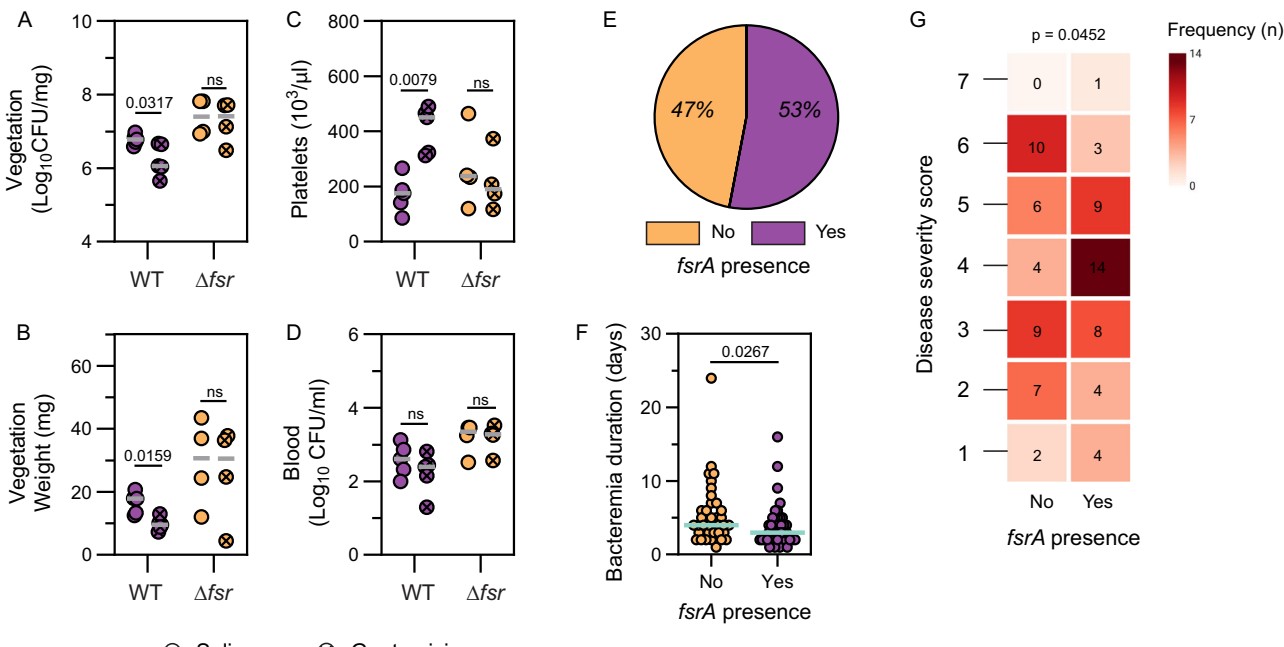

**Fig. 7 | Absence of Fsr correlates with increased antibiotic tolerance, longer bacteremia, and cases of high disease severity in IE. A-D.** Vegetation CFU (**A**), vegetation weight (**B**), platelet blood concentration (**C**), blood CFU (**D**) at 72 hpi, with and without gentamicin treatment. Median shown for $n = 5$ (WT) and $n = 4$ ($\Delta fsr$), N = 1. Statistical significance was determined with a two-tailed Mann-Whitney test between treated and untreated groups infected with the same strain. **E** *fsrA* presence (%) in *E. faecalis* isolates from IE patients. **F** Bacteremia duration of IE patients in relation to *fsrA* presence. Median shown; N = 38 (*fsrA⁻*) and N = 42 (*fsrA⁺*).

Statistical significance was determined with a Mann-Whitney test with continuity correction. **G** Cumulative disease severity score (Y-axis) of IE patients infected in relation to *fsrA* presence (X-axis). Frequency (*n*) for each disease score is shown within each heatmap box. Statistical significance was determined with Fisher's exact test; *n* = animals per group, N = independent experiments, *ns* not significant ($p \geq 0.05$). Exact *p* values are reported in the figure. Source data are provided as a Source Data file.

caspase-1 processing of pro-IL-1β (Fig. 6F). Moreover, this cleavage site was absent in rats and mice (Fig. 6F), in line with the absence of a 17 kDa fragment when rat pro-IL-1β was incubated with *E. faecalis* WT. Taken together, these findings identified a species-specific mechanism in which gelatinase activates human pro-IL-1β with potential importance for modulation of human inflammation in IE.

## Absence of Fsr correlates with increased antibiotic tolerance, longer bacteremia, and cases of high disease severity in IE

We hypothesized that the larger biofilm phenotype in $\Delta fsr$ might contribute to increased tolerance to antibiotic treatment. To test this hypothesis, we treated WT- and $\Delta fsr$-infected rats with gentamicin at 48 hpi and harvested the vegetations at 72 hpi. Vegetations presented decreased CFU and weight in the antibiotic-treated WT-infected group (Fig. 7A, B, $p = 0.0317$ & $p = 0.0159$ respectively). This decrease correlated with an increase in systemic platelet counts (Fig. 7C, $p = 0.0079$), suggesting a reduction in disease severity[69–71]. By contrast, antibiotic treatment had no effect on vegetation weight and CFU, or platelet counts in $\Delta fsr$-infected animals. CFU in blood remained unchanged for both strains, irrespective of antibiotic treatment (Fig. 7D). Microbroth dilution assay showed no difference in gentamicin MIC between the WT and $\Delta fsr$ strains (Table S2). Overall, these data demonstrated that the absence of QS results in a more recalcitrant infection in vivo, underscoring a role for Fsr in the antibiotic tolerance of *E. faecalis* biofilms in IE.

*E. faecalis* clinical isolates often exhibit a chromosomal deletion encompassing the *fsr* locus, resulting in the lack of gelatinase activity[12,13,72]. In a prior study of 80 IE isolates, 38.7 % showed an absence of gelatinase activity, with 30 % of these lacking the *fsrB* gene[13]. Although the presence of the *fsr* locus was not associated with disease causation[13], its impact on disease severity was not studied. Based on

our pre-clinical findings linking *fsr* absence to increased virulence, we hypothesized that there might be a correlation between *fsr* absence and IE severity. To address this, we examined 81 enterococcal IE cases and isolates from cohorts in the United States[73] and Switzerland with broadly similar strain diversity across cohorts based on phylogenetic analysis (Figure S5). Among the isolates, 47 % lacked *fsrA* (Fig. 7E), which consistently correlated with the absence of *fsrB*, except for isolate DVT1034, which harbored *fsrB* and a highly truncated *fsrA* gene (Figure S6). Strikingly, clinical data revealed that the absence of *fsrA* was associated with a longer duration of bacteremia (Fig. 7F, $p = 0.0267$). Although population structures and treatment regimens differed between cohorts (Supplementary Data 3), a generalized linear model analyzing bacteremia duration in *fsrA*-positive patients showed no cohort bias ($p = 0.104$). To better assess disease severity, we implemented an explorative, not clinically validated, scoring system with the objective of obtaining a multidimensional representation that would facilitate a comprehensive understanding of disease severity. The scoring system parameters included fulfillment of the modified Duke criteria for IE[74], intensive care unit (ICU) stay, length of hospital stay, need for device or valve replacement, and in-hospital mortality. The correlation of *fsrA* presence with these parameters was examined both individually and within the framework of the scoring system. Although *fsrA* presence was not directly associated with any individual parameter (Figure S7, Supplementary Data 3), we identified a significant association between high disease severity scores and the absence of *fsrA* (Fig. 7G, $p = 0.0452$).

To assess potential in-host heterogeneity for *fsrA* presence, we analyzed isolates collected at different time points from the same patients. All retained the same *fsrA* genotype over time (Table S3), suggesting that *fsr* deletion does not arise during IE. Next, we assessed whether virulence factors that anticorrelate with *fsrA* presence could

account for the prolonged bacteremia and high disease severity scores. Across all virulence factors tested (Figure S8), we found that *fsrA* was significantly anticorrelated with *esp* ($p = 4.9e^{-7}$), *EFO149* ($p = 0.0074$), *cylB* ($p = 0.01$), *cylL* ($p = 0.03$), *cylS* ($p = 0.03$). However, none of these genes were significantly associated with prolonged bacteremia or high disease severity (Figure S9), suggesting that the absence of the *fsr* locus itself is the primary driver of the observed clinical phenotypes. Overall, these findings link loss of the *fsr* locus in clinical isolates to more severe manifestations of IE in human patients, in line with our experimental observations in the rat model.

## Discussion

This work uncovers how biofilm formation is driven in enterococcal IE, redefining the role of QS as a negative regulator in IE pathogenesis. We show that fluid flow suppresses Fsr activation, demonstrating that QS is not solely determined by cell density but is also spatially regulated within vegetations. In early IE, bacteria are found within small, superficial aggregates where Fsr is inactive and thus dispensable, whereas in late IE, larger microcolonies become embedded and shielded from flow, allowing Fsr activation and making QS important for biofilm development. In contrast to the long-standing paradigm from in vitro studies in which Fsr is viewed as a positive driver of biofilm formation, we demonstrate Fsr as a negative regulator of biofilm growth in vivo, in part through its control of *gelE* and *sprE*. Transcriptomic analysis further expanded the role of Fsr beyond its known regulon, revealing that *lrgAB* is upregulated in the absence of *fsr* and potentially enhances pyruvate utilization and biofilm growth in vegetations. We also found that gelatinase can activate human pro-IL-1β, suggesting a potential species-specific mechanism for inflammation modulation by QS in humans. Finally, we show that *fsr* loss confers increased antibiotic tolerance in vivo and correlates with prolonged bacteremia and high disease severity in patients, linking the role of QS as a negative regulator of biofilm growth in vivo to clinical outcomes.

The Fsr QS system has been previously shown to promote biofilm formation in vitro in OG1RF and V583 strains[54,75,76] and contribute to disease pathogenesis in endophthalmitis and peritonitis models[22,77]. Rather than promoting biofilm formation, however, we show that induction of the Fsr QS system in vivo limits biofilm growth and reduces pathogenicity in rats. Moreover, we observed that the absence of *fsr* is associated with prolonged bacteremia in patients with IE. Consistent with our findings, the homologous Agr QS system in *S. aureus* also limits biofilm growth and promotes dispersal when activated compared to its inactive state[4,78–81]. These contrasting observations in *E. faecalis* likely arise from the dynamic and heterogeneous vegetation microenvironment, which is markedly different from the in vitro static conditions commonly used to assess biofilm formation. Besides the heart valve niche, fluid flow is expected to significantly impact QS in other hydrodynamically challenged host niches, including the bladder during intermittent urine expulsion and slow peristalsis in the gut during colonization and disease. At the same time, these niches exhibit complex topographies that could shield biofilm from flow and facilitate localized accumulation of autoinducers. Such spatial effects, previously shown for *S. aureus* and *V. cholerae* biofilms in vitro using microfluidics[4], could activate QS only in specific regions. Given the direct exposure to blood flow on the endocardial vegetation surface, we propose that adopting a QS-OFF state might be advantageous for *E. faecalis* to rapidly grow into a biofilm and withstand shear stress.

The intact *fsr* locus is variably present among *E. faecalis* strains isolated from IE, blood, urine, and feces[12,13]. A prior study reported that *fsr* was not more common in disease-associated isolates than in those colonizing healthy individuals[13]. However, its potential role in disease severity has not been previously investigated. Here, we show that the absence of the *fsr* locus correlates with larger biofilms and greater antibiotic tolerance in vivo and is also associated with prolonged

bacteremia. *S. aureus* Agr-negative isolates have also been associated with persistent bacteremia and higher mortality[82,83], often displaying a fitness advantage in the presence of antibiotics in vitro[83]. Unlike frameshift mutations in the *agr* locus of *S. aureus*[84], loss of *fsr* is typically associated with the deletion of the *fsrC*-EF_1841 region[72]. *E. faecalis* strains lacking this region harbor a highly conserved ~600 bp junction sequence, suggesting that the region spanning the deletion might be a product of horizontal gene transfer[12,72]. With the exception of isolate DVT1034, which harbored a highly truncated *fsrA* gene, the remaining *fsr*-negative strains in our study lacked both *fsrA* and *fsrB*, and exhibited absence or partial presence of *fsrC*, presumably attributed to the larger genomic region deletion previously described. What drives the loss of *fsr* in *E. faecalis* and whether *fsr*-negative strains initiate IE infection or emerge through in-host adaptation, remains unclear. Our analysis of isolates collected at multiple time points from the same patients revealed consistent *fsrA* genotypes, suggesting that *fsr*-negative strains may initiate infection rather than arise during its progression. However, as only single colonies per time point were analyzed, we cannot rule out the presence of low-frequency subpopulations with different genotypes. Thus, while our data do not support *fsr* loss arising during infection, the possibility of in-host adaptation favoring QS-deficient cheaters remains open.

IL-1β is a potent pro-inflammatory cytokine typically cleaved and activated intracellularly by caspase-1 upon induction of pyroptosis[85], but can also be activated by neutrophil elastase and cathepsin G[86]. In addition to host activation, IL-1β can also be activated by secreted bacterial proteases such as *Streptococcus pyogenes* SpeB and *Pseudomonas aeruginosa* LasB, and IL-1β cleavage has been suggested to serve as a sensor for microbial proteases[66,67]. Here, we show that *E. faecalis* gelatinase proteolytically activates human pro-IL-1β but degrades rat pro-IL-1β. We suggest that pro-IL-1β becomes accessible to biofilm-secreted gelatinase within vegetations through release from neutrophils undergoing NETosis. Although gelatinase may activate IL-1β in this context, host proteases present in vegetations are also likely to contribute to IL-1β activation[87]. Rather than serving as the sole activator, we propose that gelatinase activation of IL-1β could complement existing host activation mechanisms, thereby amplifying spatially restricted inflammation around the vegetation.

In addition to its role in regulating biofilm via *gelE* and *sprE*, we show that loss of Fsr QS also leads to a large-scale transcriptional reprogramming in *E. faecalis*, suggestive of a shift in metabolic capacity and a much larger (albeit likely indirect) *fsr* regulon than previously appreciated. Genes *lrgA* encoding a murein hydrolase regulator and *lrgB* encoding an anti-holin were the most significantly upregulated genes within Δ*fsr* vegetations. We found that *lrgA* and *lrgB* are important for *E. faecalis* to grow when pyruvate was the main carbohydrate source under both aerobic and anaerobic conditions in vitro. Moreover, our findings suggest that *lrgAB* may contribute not only to pyruvate uptake but also to its downstream catabolism. This expands on previous studies in *B. subtilis*, *S. aureus*, and *S. mutans*, where *lrgAB* homologs were primarily implicated in pyruvate transport[55–58]. However, since the *lrgA* mutant retained higher extracellular pyruvate, we cannot exclude the possibility that some of the elevated intracellular signal reflects carryover from the culture medium despite extensive PBS washes. Importantly, deletion of *lrgAB* in the Δ*fsr* background restored vegetation weight and CFU to WT levels, indicating that *lrgAB* is essential for the enhanced biofilm growth observed in the absence of *fsr*. Loss of the *fsr* locus in *E. faecalis* also resulted in the upregulation of several PTS in IE vegetations. Since *E. faecalis* is non-motile, upregulation of PTS might enhance the utilization of nutrients available in its immediate microenvironment. PTS can also regulate numerous cellular processes, including antibiotic tolerance, through the direct phosphorylation of target proteins or by interacting with them in a phosphorylation-dependent manner[88,89]. Moreover, 49 of the genes we identified as negatively regulated by Fsr overlapped with those

described in a previous study comparing in vitro grown *E. faecalis* WT and a fsrB mutant using microarrays[90], including components of the arginine deiminase pathway and other metabolic functions important for adaptation to anaerobic conditions. Interestingly, *lrgAB* was not identified as differentially expressed, suggesting that certain aspects of fsr-dependent regulation may be specific to the host infection environment. Taken together, these data suggest that the metabolic rewiring observed in the absence of *fsr* may contribute to the enhanced growth of *E. faecalis* biofilms observed in IE.

Overall, our findings show that the host microenvironment in IE exerts spatiotemporal control on the Fsr QS system, driving its activation in late IE to restrict biofilm expansion. Consequently, spontaneous deletion of the *fsr* locus, as seen in clinical isolates, or possible downregulation of *fsr* expression during infection, improves metabolic efficiency and supports biofilm expansion within the vegetation. Although QS inhibition is pursued as an anti-virulence strategy for a number of pathogens[91], our data suggest that suppressing *fsr* may not be beneficial in IE. Additional studies are warranted to evaluate whether the absence of the *fsr* locus or gelatinase function can serve as a prognostic marker for disease severity, potentially guiding more personalized treatment approaches in patients with IE. Collectively, these findings offer a deeper understanding of the infection-associated biofilm physiology of *E. faecalis* and its role in antibiotic tolerance and immune evasion, which could provide insight into other biofilm-related infections and inform future strategies for their treatment.

# Methods

### Ethics statements

Clinical data and 24 *E. faecalis* isolates from IE patients were collected in accordance with the ethical guidelines of the independent ethics committee Zurich, Switzerland. The study was approved under BASEC ID 2017-01140 and 2017-02225. Informed consent was obtained from all participants prior to data collection. Patients were eligible for inclusion if they were 18 years of age or older, had a diagnosis of IE or a high clinical suspicion of IE, and provided written informed consent after receiving information about the study. For patients temporarily unable to provide consent due to transitory mental disorders or transitory incapacity, an independent physician not involved in the study was appointed to represent the patient. Consent was then obtained from the patient as soon as they were able. If consent was subsequently denied, all collected samples were destroyed. In cases of long-standing inability to sign because of the health status, we obtained the alleged will of the patient asking the relatives or the legal spokesperson and/or legal guardian, who were then asked to provide written consent on the patient's behalf. The patient was approached for personal consent as soon as possible, and any refusal resulted in the destruction of previously collected samples. For patients with permanent incapacity, informed consent was obtained from the person authorized to act as substitute for the permanent representation (ZGB Art. 378). Patients were excluded if participation was contraindicated on ethical grounds, including advanced dementia or an inability to provide informed consent due to language barriers. Previously published clinical data and genomic information from 57 *E. faecalis* isolates collected from patients with definite or probable IE at the University of Pittsburgh Medical Center were also included[73]. This study was approved with a waiver of informed consent by the Institutional Review Board at the University of Pittsburgh (protocol no. STUDY22050046). Male Sprague-Dawley rats (7–9 weeks old) were used in accordance with the NTU Institutional Animal Care and Use Committee guidelines (Animal utilization protocols A19091 and A24076).

### Materials

All reagents and instruments used for RNA isolation, SDS-PAGE, and western blotting were purchased from Thermo Fisher Scientific, unless otherwise specified. Antibodies and respective dilutions used for immunofluorescence (IF) and western blotting (WB) in this study include rabbit anti-Streptococcus Group D (Antigen D) (1:500 for IF, Cat. Nr. 12-6231D, American Research Products, Inc), rabbit anti-histone H3 (citrulline R2 + R8 + R17) (1:500 for IF, Cat. Nr. ab5103, abcam), mouse anti-myeloperoxidase (1:50 for IF, Cat. Nr. NBP1-51148, Novus Biologicals), rabbit anti-IL-1β (1:50 for IF and 1:1000 for WB, Cat. Nr. ab283818, abcam), rabbit anti-gelE (1:500 for WB, Cat. Nr. PA5-117682, Invitrogen), rabbit anti-beta-actin (1:4000 for WB, Cat. Nr. ab8227, abcam), goat anti-Rabbit IgG (H + L) HRP-conjugated antibody (1:5000 for WB, Cat. Nr. 31460, Invitrogen), goat anti-rabbit IgG Alexa 488 (1:1000 for IF, Cat. Nr. A11034, Invitrogen), goat anti-mouse IgG1 Alexa 633 (1:1000 for IF, Cat. Nr. A-21126, Invitrogen).

### Bacterial cultures

Strains used in in vitro and in vivo assays are summarized in Supplementary Data 5. All strains were routinely grown in brain heart infusion (BHI) broth at 37 °C without shaking, in ambient air. For strains transformed with P*gelE*-GFP and P*cfb*-GFP, BHI was supplemented with spectinomycin (120 μg/ml). Growth curves were generated by diluting overnight cultures 1:100 in fresh BHI supplemented with 40 % horse serum (Sigma-Aldrich) (BHIS) and adding 100 μl of the diluted cultures in triplicate to a 96-well plate. The plate was then incubated for 18 h at 37 °C in a plate reader (TECAN), with absorbance recorded at 600 nm every 20 min in ambient air conditions without shaking. To validate P*gelE*-GFP and P*cfb*-GFP expression in WT and Δ*fsr* strains, overnight cultures were diluted 1:100 in M1 media supplemented with glucose (110 mM) selected for its low autofluorescence and incubated for 18 h at 37 °C without shaking. Absorbance at 600 nm and fluorescence (ex 488 nm/em 522 nm) were measured every 20 min. To investigate the role of *lrgAB*, overnight cultures were resuspended 1:100 in M1 media[92] supplemented with pyruvate (110 mM) or glucose (110 mM) or resuspended in BHI or BHI with 0.4 % Triton X-100 and incubated in ambient air or anaerobic conditions for 24 h at 37 °C without shaking with absorbance recorded at 600 nm every 20 min. The 57 *E. faecalis* isolates collected from patients with definite or probable IE at University of Pittsburgh Medical Center strains were assessed for gelatinase activity on gelatin agar plates in a separate study[93]. All *fsrA*-positive strains exhibited gelatinase activity, except for DVT988 and DVT1721 strains. None of the *fsrA*-negative strains exhibited gelatinase activity.

### Flow rate calculation

Shear stress on the aortic valves at a normal resting heart rate ranges from 10 to 28.5 dynes/cm², based on experimental and computational data[41–43]. To mimic the microenvironment of the aortic valves in our in vitro assays, we exposed bacteria to 20 dynes/cm², approximating the mean value of this range. To achieve 20 dynes/cm² in microfluidic channels (μ-Slide VI 0.4, ibidi), we calculated the required flow rate using the following formula provided by the manufacturer:

$\tau = \eta \cdot 176.1 \cdot \Phi$ [mL/min], where $\tau$ = shear stress (dynes/cm²), $\eta$ = dynamical viscosity (dynes × s/cm²) and $\Phi$ = flow rate. Dynamic viscosity of BHIS media is 0.0101171 dynes × s/cm² at 37 °C, as determined using a viscometer. Thus, the required flow rate to achieve 20 dynes/cm² is 11.226 mL/min.

### Incubation of bacteria under fluid flow conditions

To assess the early impact of fluid flow on *E. faecalis*, 30 μl of stationary-phase bacterial cultures of OG1RF WT were seeded into microfluidic channels with uncoated polymer surface (μ-Slide VI 0.4, ibidi) and incubated for 30 min at 37 °C to allow bacterial attachment. After the 30-minute incubation, media in the microchannels was exchanged with fresh BHIS to remove non-attached bacteria, as BHIS approximates in vivo growth conditions[94,95]. For flow conditions, bacteria were exposed to BHIS at defined shear stress levels: low

(1 dynes/cm$^2$), intermediate (10 dynes/cm$^2$), and high shear stress (20 dynes/cm$^2$), the latter corresponding to the aortic valve shear stress levels. Pulsatile flow was applied using a peristaltic pump (IPC 12 model, ISMATEC), programmed to mimic cardiac diastole and systole by alternating between pauses of 0.5 s and BHIS delivery for 0.3 s for a total duration of 30 min at 37 °C. For static conditions, bacteria were incubated inside the microchannels for 30 min at 37 °C without fluid flow. To harvest RNA for sequencing and qPCR after the incubation period, BHIS in the microchannels was replaced immediately with RNAprotect Bacteria reagent (Qiagen), followed by incubation for 5 min at room temperature (RT). Bacteria were then detached from the microchannels using a 1 mL syringe prefilled with 200 μL of RNAprotect by applying rapid plunging movements to shear bacteria from the surface. Detached bacteria were pelleted at 8000 g for 15 min at RT and stored in -80 °C until RNA isolation.

## RNA isolation

Bacterial samples harvested from microfluidic channels and treated with RNAprotect or vegetations stored in RNALater Stabilization Solution were transferred in 1 mL Trizol and homogenized with lysing matrix B (MP Biomedicals) at 6 m/s for 3 cycles of 40 s with ice cooling between cycles. Chloroform was added to Trizol at a 1:5 ratio, mixed by gentle inversion, and incubated for 2 min at RT. Following centrifugation at 12,000 g for 15 min at 4 °C, the aqueous phase was mixed with an equal volume of 80% ethanol. Samples were loaded onto a RNeasy Mini spin column (Qiagen), and total RNA was purified according to the manufacturer's instructions, including an on-column DNase treatment step. RNA quantity was measured using the Qubit RNA Broad Range Assay Kit, and genomic DNA contamination was assessed using the Qubit dsDNA HS Assay Kit. RNA quality was evaluated using the RNA ScreenTape and RNA Sample Buffer on a TapeStation 2200 instrument (Agilent), following the manufacturer's instructions.

## Transcriptomic sequencing analysis

For bacterial total RNA samples isolated from fluid flow conditions, library preparation was performed using the TruSeq Stranded mRNA Library Prep Kit (Illumina). The libraries from fluid flow experiments were sequenced on an Illumina HiSeq 2500 platform to generate 75 bp paired-end reads. Vegetation total RNA samples were processed using the Ribo-Zero Plus rRNA depletion kit (Illumina, USA) to remove rat and bacterial rRNA, followed by library preparation with the TruSeq Stranded mRNA Library Prep Kit (Illumina). These libraries were sequenced on an Illumina Illumina HiSeqX v2.5 platform to generate 150 bp paired-end reads. All data were analyzed using a pipeline on the Galaxy platform[96]. Briefly, quality control of the sequencing data was performed using FastQC, followed by filtering with SortMeRNA (Version 4.3.6) to remove rRNA sequences. Reads were aligned to the *E. faecalis* OG1RF reference genome (Accession number: GCF_000172575.2_ASM17257v2) using Bowtie2 (Version 2.5.3)[97] and reads per gene were quantified using HTSeq-count (Version 2.0.5). Differential expression analysis was conducted with edgeR (Version 3.36.0) with statistical significance set at FDR < 0.05. GO enrichment analysis was performed using goseq (Version 1.50.0).

## Quantitative PCR

cDNA was synthesized from equal amounts of total RNA from each sample using the SuperScript III First-Strand Synthesis SuperMix (Thermo Fisher Scientific) according to the manufacturer's instructions, including a minus reverse transcriptase control for each sample. qPCR primer efficiency was determined by performing qPCRs on 10-fold serial dilutions of pooled cDNA (1:2 to 1:2000) and calculating efficiency (%) using StepOne software (version 2.3, Applied Biosystems). Primer pairs with efficiency 90-100% were selected for subsequent qPCR experiments. All qPCR reactions were performed in duplicates, including minus reverse transcriptase and no-template controls, using the KAPA SYBR FAST qPCR Master Mix (2X) kit (KAPA BIOSYSTEMS) on a StepOnePlus Real-Time PCR System (Applied Biosystems). Data were analyzed using the $2^{-\Delta\Delta Ct}$ method and presented either as ΔCt values or as log2 fold change (log2FC) relative to the control group. To normalize gene expression in bacterial samples exposed to fluid flow, the geometric mean of *recA* and *dnaB* was used, based on their stable expression under both static and fluid flow conditions as indicated by transcriptomic data and validation using the Bestkeeper v1 analysis tool[98]. To normalize gene expression in vegetation samples, *recA* was chosen as the reference gene due to its stable expression both in vivo and in vitro, as demonstrated by prior transcriptomic data. Primers used are summarized in Supplementary Data 6.

## DNA manipulation and construction of deletion mutants

Generation of *E. faecalis* knockout mutants was done by allelic replacement using a temperature-sensitive shuttle vector previously described[99]. Vector pGCP213 was linearized using appropriate restriction enzymes (New England Biolabs, USA) for the construction of the deletion mutants. The flanking regions were fused together by overlap extension PCR to generate the final insert. Linearized vector and inserts were then ligated using the T4 ligase or In-Fusion HD cloning kit (Clontech, TaKaRa, Japan) and transformed into Stellar competent cells. Successful plasmid constructs were verified by Sanger sequencing and subsequently extracted and transformed into OG1RF. Transformants were selected with erythromycin (25 μg/mL) at 30 °C and then passaged at a nonpermissive temperature at 42 °C with erythromycin to select for bacteria with successful plasmid integration into the chromosome. For plasmid excision, bacteria were serially passaged at 37 °C without erythromycin for erythromycin-sensitive colonies. These colonies were then subjected to PCR screening to identify successful deletion mutants. All primers used are listed in Supplementary Data 6. The Δ*fsr* strain, as well as the previously constructed Δ*gelE* strain[100], did not show any additional deletions or major mutations compared to WT based on assessment with whole genome sequencing.

The proteolytically inactive GelE strain was generated by chromosomal insertion of gelE*(E352A), which has an E352A point mutation in the active site[101] and an A29 silent mutation to introduce a SacII restriction site as a unique identifier of engineered GelE strains[102], into *E. faecalis* OG1RF Δ*gelE*[103]. The silent mutation was first incorporated by amplifying WT gelE sequence from OG1RF genomic DNA, including -0.5 kb upstream and downstream regions and 15 bp overhangs homologous to vector ends, using the primer pairs gelE_ins_1/gelE_ins_2 and gelE_ins_3/gelE_ins_4, which were subsequently fused by overlap extension PCR to generate the final insert. pGCP213 was linearized by inverse PCR using pGCP213_F and pGCP213_R. The insert and linearized pGCP213 were then ligated by In-Fusion HD Cloning Kit (Clontech, TaKaRa, Japan) and transformed to *E. coli* Stellar cells as described above, generating pGCP213::*gelE**. pGCP213::*gelE** was then subjected to site directed mutagenesis (SDM) by PCR using overlapping mutagenic primers gelE_E352A_F/gelE_E352A_R. The PCR product was treated with DpnI (New England Biolabs, USA), purified and re-ligated with In-Fusion HD Cloning Kit, generating pGCP213::*gelE* (E352A). The final plasmid was similarly propagated in *E. coli* Stellar cells and subsequently electroporated into Δ*gelE* OG1RF for serial passaging and chromosomal integration. Secretion of inactive gelatinase was validated by western blot of cell-free culture supernatants and gelatinase assay on Todd-Hewitt agar + 3% gelatine[44].

For the plasmid complementation of GelE, promoter-free *gelE* sequence from wildtype OG1RF was amplified with primers gelE_pTCV-Ptet_F/gelE_pTCV-Ptet_R (Supplementary Data 6). The *gelE* insert is ligated to a BamHI/PstI-digested pTCV-P$_{tet}$ shuttle vector using InFusion HD Cloning Kit (ClonTech, TaKaRa, Japan) and transformed into

Stellar competent cells, generating pTCV-P$_{tet}$::*gelE*. Verified pTCV-P$_{tet}$::*gelE* constructs were extracted, transformed into *E. faecalis* OG1RF Δ*gelE* via electroporation and selected on BHI agar with 25 µg/mL erythromycin. Correct transformants were subsequently maintained in erythromycin (25 µg/mL) for complementation experiments.

To construct the P*gelE*-GFP reporter plasmid, P*cfb*-GFP was linearized by inverse PCR with primers pBSU101_Pcfb_F/pBSU101_Pcfb_R (Supplementary Data 6) to exclude the constitutive *cfb* promoter. The *gelE* promoter region[47] including the ribosomal binding site (positions -86 to +121 bp relative to +1 transcriptional start site), was amplified from wildtype OG1RF with primers Pgele_F/PgelE_R (Supplementary Data 6). The *gelE* promoter was then ligated to the linearized vector using InFusion HD Cloning Kit (ClonTech, TaKaRa, Japan) and transformed into Stellar competent cells, generating P*gelE*-GFP. Both P*gelE*-GFP and P*cfb*-GFP were transformed into wildtype and Δ*fsr* OG1RF via electroporation and selected on BHI agar with 120 µg/mL spectinomycin. For in vitro experiments, *E. faecalis* containing P*gelE*-GFP and P*cfb*-GFP were maintained in 120 µg/mL spectinomycin.

### Infective Endocarditis rat model

Male Sprague-Dawley rats (7-9 weeks old) were kept in cages enriched with bedding and wooden blocks. Food and water were provided ad libitum. Infective endocarditis was established as previously described with minor modifications[104]. Briefly, polytetrafluoroethylene (PTFE) catheters (0.6 mm outer diameter, 6.5 cm in length), with a silicon ring positioned at 5 cm, were sealed at both ends with 28-gauge stainless steel plugs (Braintree Scientific) and sterilized by autoclaving. Aortic valve lesions were induced by catheterizing the left ventricle via the right carotid artery of isoflurane-anesthetized rats. Proper placement of the catheter was assessed by momentary disturbance of the blood flow pattern monitored by a rat paw pulse oximeter (Physiosuite, Kent Scientific) and by the vigorous pulsation of the catheter once inserted in the left ventricle. Catheter was secured in place by tying a silk suture (size 7-0) around the carotid artery and the rostral end of the silicon ring. IE was induced by injecting a 200-µL bacterial suspension ($2 \times 10^6$ CFU) in PBS via the dorsal penile vein at 24 h post-catheterization. To test antibiotic efficacy, gentamicin (3 mg/kg of body weight) or saline were injected via the dorsal penile vein at 48 hpi. To maintain plasmid stability in WT P*cfb*-GFP and WT P*gelE*-GFP strains for 24 h in vivo, spectinomycin (20 mg/kg of body weight) was injected via the dorsal penile vein at 0 hpi. Analgesia was provided daily by administering buprenorphine (0.5 mg/kg) subcutaneously. At 6, 24, or 72 hpi, rats were anesthetized with isoflurane inhalation and sacrificed in a $CO_2$ chamber, followed by cervical dislocation. The heart was extracted and catheter placement inside the left ventricle was verified. Animals with incorrect catheter placement were excluded from downstream analysis.

For CFU quantification in tissue samples, valvular vegetations, liver, and spleen were weighed and homogenized in a lysing matrix M tube (Cat. Nr. 116923050-CF, MP Biomedicals) with PBS using a FastPrep24 instrument (3 cycles at 4 m/s for 20 s each). Homogenates were serially diluted, plated using the drop plate method on BHI agar with and without rifampicin (25 µg/mL), or BHI agar with and without spectinomycin (120 µg/mL), and incubated for 18 h at 37 °C.

Blood samples were collected via heart puncture under isoflurane anesthesia, followed by euthanasia in $CO_2$ chamber and cervical dislocation. Blood was transferred in EDTA-coated tubes to prevent coagulation and analyzed using a XN-10 Automated Hematology Analyzer (SYSMEX) at the Advanced Molecular Pathology Laboratory (AMPL) at the Agency for Science, Technology and Research, Singapore. For CFU quantification, 100-500 µL of blood were plated on BHI agar with and without rifampicin (25 µg/mL).

### Tissue immunofluorescence staining

Aortic valve vegetations for immunofluorescence staining were routinely collected and fixed in 4 % paraformaldehyde for 24 h at 4 °C,

followed by sequential cryoprotection in 15 and 30 % sucrose in PBS at 4 °C until the samples sank. Samples were embedded in optimal cutting temperature (OCT) compound (Tissue-Tek, SAKURA Finetek), snap-frozen in a dry ice isopropanol bath, and sectioned at a thickness of 5-8 µm using a cryostat (CM1950, Leica). Vegetations intended for IL-1β staining and vegetations infected with WT P*cfb*-GFP and WT P*gelE*-GFP strains were embedded in OCT and snap-frozen immediately after harvesting to prevent antigen masking and GFP quenching respectively, then sectioned at a thickness of 7 µm and fixed in 4 % paraformaldehyde for 10 min at 4 °C. All sections were permeabilized in 0.5 % Triton X-100 for 5 min and blocked with buffer containing 5 % goat serum, 5 % bovine serum albumin (BSA), and 0.05 % Triton X-100 in PBS for 30 min at RT. Sections were incubated overnight at 4 °C with primary antibodies diluted in 1 % goat serum, 1 % BSA, and 0.05 % Triton X-100 in PBS. After washing 3 times with PBS, sections were incubated with fluorophore-conjugated secondary antibodies for 1 h at RT in the dark. Nuclei were counterstained with 4',6-diamidino-2-phenylindole (DAPI) (1:500, stock concentration 5 mg/mL, Thermo Fisher Scientific) for 5 min at RT. Sections were mounted with antifade mounting medium and stored in -20 °C until microscopy imaging. Secondary-only antibody controls for goat anti-rabbit IgG Alexa 488 (Cat. Nr. A11034, Invitrogen) and goat anti-mouse IgG1 Alexa 633 (1:1000, Cat. Nr. A-21126, Invitrogen) are shown in Figure S10A, B. Antibodies used are listed in the Materials subsection. All microscopy images were acquired at NTU Optical Bio-Imaging Center (NOBIC) imaging facilities at SCELSE.

### Microscopy and image analysis

To obtain images of whole vegetation cross-sections, tile epifluorescent images were captured on a Carl Zeiss Axio Observer Z1 inverted microscope equipped with a 20x or 40x objective using ZEN 2 (blue) software (Zeiss). Snapshots or Z-stacks were acquired on a Carl Zeiss LSM 780 laser scanning confocal using ZEN 2.3 SP1 FP3 (black) software (Zeiss). Imaging parameters were kept constant across each experiment. Maximum intensity projections were generated from Z-stacks using ImageJ (version 1.54 f). Linear brightness and contrast adjustments were applied for visualization purposes where necessary using ImageJ.

Tile epifluorescent images of whole vegetation cross-sections were thresholded to determine the biofilm and vegetation outlines based on antigen D and DAPI fluorescent signals respectively. Based on these outlines, the total biofilm and vegetation area was measured using ImageJ and biofilm area (%) was calculated using the formula: (biofilm area/total vegetation area) x 100. For each independent vegetation sample, biofilm area (%) was averaged from 3 cross-sections 80 µm apart, which served as technical replicates. Individual biofilm microcolony area was quantified using ImageJ in snapshots obtained at 63x with LSCM across 3 cross-sections spaced 80 µm apart, from 2–4 vegetations per condition. A total of 20 - 65 microcolonies were quantified per vegetation. For Fig. 1D-F and Figure S1K, microcolony area, distance from vegetation surface, and gelatinase expression were quantified in ImageJ using tile images acquired at 40x with an epi-fluorescence inverted microscope, across 2 or more cross-sections spaced 80 µm apart, from 2 - 3 vegetations per condition. Microcolony contours were defined based on AgD staining and used to measure area. The same contours were applied to the GFP channel to calculate mean fluorescence intensity. Distance from the vegetation surface was measured by drawing a straight line from the center of the microcolony to the nearest vegetation boundary. Microcolonies that were partly or fully exposed at the vegetation surface were classified as surface microcolonies. To classify microcolonies as small or large, we used the median area (36.62 µm²) of all microcolonies in the dataset as the cutoff between the two groups.

### Western blotting

Aortic valve vegetations were homogenized using a combination of lysing matrix M and B (MP Biomedicals) to ensure lysis of both rat

and bacterial cells. Vegetations were placed in tubes with the lysing matrices and filled with RIPA lysis buffer (250 μL/5 mg tissue), supplemented with 3x Halt Protease Inhibitor Cocktail and 1x EDTA. Homogenization was performed using a FastPrep24 instrument (MP Biomedicals) for 4 rounds at 4 m/s, 20 s each. Samples were then incubated on ice for 30 min and sonicated for 3 rounds (15 s on, 5 s off) on ice at 20 % power using a sonicator (Vibra cell, SONICS). Lysates were centrifuged at $10,000 \times g$ for 20 min at 4 °C, after which the supernatant with the soluble proteins was collected. Total protein concentration was measured using the Pierce BCA protein assay kit. 25 or 50 μg of vegetation protein for detecting rat only or rat and bacterial targets respectively, and 10 μL of supernatant from IL-1β cleaving assays were separated by SDS-PAGE on NuPAGE 4-12% Bis-Tris gels in ice-cold 1x MES or 1x MOPS SDS Running Buffer. Electrophoresis was performed at 120 V for 40-60 min. Proteins were transferred to polyvinylidene fluoride (PVDF) membranes using the iBlot 2 Gel Transfer Device. Membranes were blocked with 5 % BSA in 1 X Tris-Buffered Saline with 0.1 % Tween 20 (TBST) for 1 h at RT with shaking. Membranes were incubated with primary antibodies diluted in 1 % BSA in TBST overnight at 4 °C with gentle shaking. Membranes were washed 3 times in TBST and incubated with HRP-conjugated secondary antibody diluted in 1 % BSA in TBST for 1 h at RT. Protein bands were visualized using SuperSignal West Femto Maximum Sensitivity Substrate and imaged in Amersham ImageQuant 800. Loading controls were run on the same blot. For reprobing, membrane-bound primary and secondary antibodies were removed using Restore PLUS Western Blot Stripping Buffer by incubating the membrane for 30 min at 50 °C with shaking. Protein band density was quantified using ImageJ. Antibodies used are listed in the Materials subsection. Uncropped and unprocessed scans for blots in main figures are available in the Source Data file and for blots in Supplementary Figs. are available in Supplementary Information.

### Flow cytometry

Aortic valve vegetations were excised, weighed, and minced in 500 μL of digestion buffer (0.1 mg/mL Liberase in PBS). Samples were incubated at 37 °C for 30 min with gentle rocking. After incubation, 500 μL of flow cytometry staining buffer (FCSB; 1 % BSA and 0.1% sodium azide in PBS) was added, and the supernatants were filtered through a 35-μm cell strainer into a 50 mL tube. An additional 500 μL of FCSB was used to flush the strainer, bringing the total volume to 1.5 mL. Filtrates were centrifuged at $500 \times g$ for 5 min at 4 °C, and cell pellets were resuspended in 200 μL of FCSB. 100 μL per sample were subsequently used for each staining condition. Non-specific Fc receptor binding was blocked by incubating with mouse anti-CD32 (1:100, Cat. Nr. 550270, BD Pharmingen) for 20 min at 4 °C. Following blocking, samples were incubated with mouse anti-CD45:Alexa Fluor 700 (1:10, Cat. Nr. MCA43A700, Biorad) and mouse anti-RP-1:Alexa Fluor 647 (1:20, Cat. Nr. 550000, BD Pharmingen) for 30 min at 4 °C in the dark. After incubation, cells were washed and fixed in 4 % PFA for 15 min at 4 °C in the dark, followed by washing and resuspension in 100 μL FCSB. To obtain absolute cell counts, 100 μL of AccuCheck Counting Beads (Thermo Fisher Scientific) were added to each sample by reverse pipetting. Stained cells were acquired using a BD LSRFortessa X-20 flow cytometer equipped with 5 lasers (488 nm, 535 nm, 633 nm, 405 nm, 355 nm). A total of 10000 events were collected per sample. Compensation was applied using the AbC Anti-Mouse Bead Kit (Thermo Fisher Scientific), and fluorescence-minus-one (FMO) controls were used to set gating thresholds. Flow cytometry data were analyzed using FlowJo (version 10.7.1) with gating strategy shown in Figure S10C–H. Absolute cell counts were calculated according to the manufacturer's instructions for AccuCheck Counting Beads and normalized to total vegetation weight.

### IL-1β cleaving assay

*E. faecalis* overnight cultures of OG1RF WT and mutants were diluted 1:10 in 200 μl fresh BHI supplemented with human pro-IL-1β (100 ng/mL; Cat. Nr. 10139-H07E, Sino Biological) and incubated for 2, 4, 6 and 24 h, or with rat pro-IL-1β (10 ng/ml, Cat. Nr. 80023-R07E, Sino Biological) and incubated at 2, 4, and 6 h at 37 °C, static conditions. Media was additionally supplemented with 25 μg/ml erythromycin when Δ*gelE* pTCV-Ptet and Δ*gelE* pTCV-Ptet-*gelE* were incubated in the presence of human pro-IL-1β (100 ng/mL) for 18 h at 37 °C, static conditions. After incubation, bacteria were pelleted, and supernatants were stored at -20 °C until separation with SDS-PAGE and western blotting. Sterile BHI media supplemented with human pro-IL-1β served as a negative control.

For mass spectrometry analysis, supernatants harvested at 0, 6 and 18 h were separated with SDS-PAGE and stained with Coomassie Blue. Protein bands corresponding to pro-IL-1β and the cleaved fragment of interest at ~17 kDa were excised and stored in miliQ water. Peptide mass spectrometry and analysis was performed at the Taplin Mass Spectrometry Facility, Harvard Medical School, Boston, Massachusetts, USA. Peptide identifications were filtered to retain only those uniquely detected at 6 and 18 h compared to 0 h and further prioritized by relative abundance ($\geq 10^6$) (Supplementary Data 4). Among the most abundant new peptides, we identified one beginning at residue 105, consistent with the generation of a ~17 kDa fragment observed by western blotting.

### Mammalian cell culture

HEK-Blue IL1R cells (invivogen), which is a HEK 293 reporter cells for human and murine IL-1α & IL-1β cytokines, were cultured in DMEM (Gibco) supplemented with 10% (v/v) heat-inactivated fetal bovine serum, 100 U/mL penicillin, 100 μg/mL streptomycin, and 1x HEK-Blue selection (invivogen) at 37 °C in a humidified incubator with 5% $CO_2$. Cells were passaged when they reached 80% confluency. For passaging or harvesting cells for assays, cells were washed and incubated in PBS for 2 min at 37 °C, followed by vigorous pipetting of PBS over the cell monolayer to detach the cells.

### IL-1 signaling assay

*E. faecalis* overnight cultures of OG1RF WT and mutants were diluted 1:10 in 1 mL fresh BHI supplemented with either human pro-IL-1β (100 pg/mL), or human IL-1β (100 pg/mL, Cat. Nr. ab9617, Abcam), or rat pro-IL-1β (10 ng/mL, Cat. Nr. 80023-R07E, Sino Biological), or rat IL-1β (10 ng/mL, Cat. Nr. ab281807, Abcam) and incubated for 6 h at 37 °C, static conditions. After incubation, bacteria were pelleted, and supernatants were filter-sterilized. 20 μl of bacterial supernatant filtrate were added to 50,000 HEK-Blue IL-1R cells (invivogen) in 180 μl DMEM supplemented with 10% FBS and 100 U/mL Penicillin-Streptomycin in a 96-well plate. As negative controls, sterile BHI, DMEM and DMEM supplemented with human pro-IL-1β (100 pg/mL) or rat pro-IL-1β (10 ng/mL) were used instead of bacterial supernatants. DMEM supplemented with human mature IL-1 (100 pg/mL) or rat mature IL-1β (10 ng/mL) served as a positive control. Samples were incubated for 18 h at 37 °C. The following day, 20 μl of HEK-Blue IL-1R cell supernatants were transferred to 180 μl of QUANTI-Blue Solution (invivogen) in a 96-well plate followed by measuring absorbance at 620 nm every 15 min in a plate reader (TECAN) at 37 °C for 2 h.

### Relative metabolite quantification by gas chromatography-mass spectrometry (GC-MS)

Bacteria were grown in M1 media supplemented with 10 mM pyruvate for 18 h at 37 °C under aerobic or anaerobic conditions. Culture supernatants were collected, and cell pellets were washed with PBS. To the cell pellet, 50 μl chloroform were added, followed by 200 μl water/methanol (1:3), containing 3 nmol $^{13}C_3/^{15}N$-β-alanine (Cambridge Isotope Laboratories, CNLM-3946) as internal standard. Similarly, 50 μl

chloroform were added to supernatant samples (50 μl), followed by 150 μl methanol, containing 4 nmol $^{13}C_3/^{15}N$-β-alanine as internal standard. Samples were mixed (30 min, 4 °C) using a sample shaker, followed by the sedimentation of insoluble material (20,000 × g, 4 °C, 6 min). Supernatants were transferred to microfuge tubes, containing 100 μl ultrapure water, resulting in a biphasic extract (chloroform/methanol/water, 1:3:3). Solutions were mixed vigorously for 1 min and the phases separated by centrifugation (20,000 × g, 4 °C, 6 min). The upper polar phases were moved to new microfuge tubes and dried sequentially in mass spectrometry vial inserts using a centrifugal evaporator. Samples were re-dried with methanol and derivatized through the addition of 25 μl pyridine (Sigma-Aldrich, 270970) and 25 μl N-tert-butyldimethylsilyl-N-methyl trifluoroacetamide (MTBSTFA) with 1% tert-butyldimethylchlorosilane (TBDMCS, Supelco, 00942). Derivatization was allowed to complete during incubation at RT for 1 h with gentle agitation prior to analysis as outlined below.

Samples were analyzed using an 8890 GC System (Agilent Technologies) equipped with a DB5 capillary column (J&W Scientific, 30 m, 250 μm inner diameter, 0.25-μm film thickness, 10-m inert duraguard) connected to a 5977B GC/MSD in electron impact (EI) mode linked to a 7693 A autosampler (Agilent Technologies). The GC–MS settings were as follows: Inlet temperature: 270 °C, MS transfer line temperature: 280 °C, MS source temperature: 230 °C and MS quadrupole temperature: 150 °C. Samples were analyzed in splitless mode, following injection of 1 μl. The oven temperature gradient during the sample run was as follows: 60 °C (2 min hold); 60 °C to 320 °C ramp at 20 °C/min, holding 320 °C for 2 min. The MS was operated in scan mode (m/z 70-700), with a solvent delay of 8 min. Peak picking and integrations were performed using MassHunter Quantitative Analysis Software 12.0 (Agilent Technologies). The area under the curve of suitable ions, corresponding to loss of the tert-butyl group (m/z 261, lactic acid-2-TBDMS, [M-57]⁺; m/z 259, pyruvic acid-2-TBDMS, [M-57]⁺; and m/z 260, L-alanine-2-TBDMS, [M-57]⁺), was determined. These ions were quantified at their peak apex, based on the analysis of authentic standards, with a retention time right and left delta of 0.08 min. The signal in each sample was normalized to that of the internal standard ($^{13}C_3/^{15}N$-β-alanine-2-TBDMS, m/z 220, corresponding to a prominent fragment containing two labeled atoms). The relative abundance was determined compared to the average abundance in the wild type control group (set to 1). Normalizations and calculations of relative abundances were performed using Excel (Microsoft). Data associated with analysis of relevant standards, total ion chromatographs, and extracted ion chromatographs are shown in Supplementary Information (Figure S11– S13).

## PCR screening of clinical isolates
The presence of fsrA was detected by PCR amplification in vitro (ENVALVE cohort, Switzerland) or using BLAST (Pittsburgh cohort). Bacteria from blood cultures at different time points within the bacteremia episode (Table S3) or heart valve samples were grown overnight on COS Agar at 37 °C. Single colonies were resuspended in DreamTaq Hot Start Green PCR Master Mix (Cat. Nr. K9021, Thermo Fisher Scientific) with primer 2 and 3 (Supplementary Data 6) at a concentration of 1.6 μM. The PCR program consisted of an initial denaturation step at 94 °C for 2 min, followed by 25 cycles of amplification (denaturation at 94 °C for 30 s, annealing at 54.1 °C for 30 s, and extension at 72 °C for 1 min) and then a final extension at 72 °C for 6 min. PCR products were analyzed by electrophoresis on a 1 % agarose gel and stained by GelRed. Amplified fragments of fsrA corresponded to 740 bp. OG1RF WT and Δfsr were included as positive and negative control respectively.

## Whole genome sequencing
Bacterial pellets from the clinical isolates of the Swiss ENVALVE cohort were resuspended in 1 mL NAP buffer and sent for sequencing at Microsynth AG (Balgach, Switzerland). DNA was extracted using ZymoResearch Quick-DNA HMW MagBead Kit, (D6060), following manufacturer's instructions and was quantified using the Pico488 dsDNA assay (Lumiprobe). Short-read Illumina DNA Tagmentation libraries were constructed using unique dual indexes to sequence the DNA according to the manufacturer's instructions and were sequenced on a NovaSeq6000 apparatus using a sequencing kit with 300 cycles in paired-end fashion.

Available raw reads of the Pittsburgh cohort (BioProject PRJNA729754) were downloaded and analyzed together with the raw reads of the Zurich cohort. Illumina reads were trimmed using fastp v0.22.0 in paired-end mode. Adapter sequences were removed using the automatic adapter detection mode. Quality-based trimming was performed with a 4 bp sliding window, cutting when the average quality within the window fell below 20. Additionally, individual bases with a Phred quality score below 30 were filtered out to ensure high base call accuracy ( > 99.9%). Reads shorter than 50 base pairs after trimming were discarded[105,106]. Read quality before and after trimming was assessed using FastQC v0.12.1[107]. Genomes were assembled de novo using SPAdes v4.1.0 without further error correction and set to 'careful'[108]. Contigs shorter than 200 bp were removed using SeqKit v2.10.0[109] and assembly quality was assessed with QUAST v5.0.2[110]. Genome annotation was performed with Prokka v1.14.6[111], with functional annotation led by protein sequences from E. faecalis strains C1336 (SAMN44727039) or OG1RF (SAMN02603002) via the –proteins option. Sequence types (ST) were assigned with the PubMLST typing schemes[112] using mlst v2.16.1 (https://github.com/tseemann/mlst). Virulence factors were identified using ABRicate v1.0.1 (https://github.com/tseemann/abricate), blasting against the virulence factor database (VFDB[113], accessed 02.06.2025).

A pangenome was defined with PIRATE v1.0.5[114] using a 98% blastp identity threshold, identifying 6193 gene families, of which 2175 core genes (present in >= 95% of isolates) were retained for phylogenetic analysis. A phylogeny, based on the core-genome alignment was created by IQ-TREE v2.2.03[115] with automatic model selection. The resulting tree was annotated and visualized using iTOL[116].

Variant calling was performed using Snippy v4.6.0 (https://github.com/tseemann/snippy), with E. faecalis strains C1336 (SAMN44727039) and OG1RF (SAMN02603002) used as references. SNPs were identified from alignments against OG1RF to examine variants in the genes fsrA and fsrB, and from alignments against C1336 to investigate variants in elrABCDER, esp, fsrC, gelE, and sprE.

## Crystal violet assay
Bacteria grown overnight in BHI were washed and diluted 1:100 in fresh BHI. 100 μL of each strain tested was transferred in triplicates in a tissue culture-treated or non-treated polystyrene 96-well plate and incubated for 24 h at 37 °C. At the incubation endpoint, wells were washed with PBS to remove non-attached bacteria and then 95 % ethanol was added for 15 min at RT to fix the biofilm. Ethanol was removed and wells were washed twice in PBS, before staining biofilm with 0.1% crystal violet for 10 min. Wells were washed 3 times with PBS, followed by addition of 95 % ethanol for 10 min to dissolve crystal violet and mixing. Absorbance at 540 nm was measured in each well using a spectrophotometer.

## Microbroth dilution assay
Bacteria grown overnight in BHI were washed and resuspended in cation-adjusted Mueller-Hinton Broth (MHB). Bacterial concentration was adjusted to $1 \times 10^6$ CFU/mL in MHB and incubated with a 2-fold serial dilution of ampicillin (0.25–16 μg/mL) and gentamicin (2–128 μg/mL) in a 96-well plate for 18 h at 37 °C without shaking. Wells without antibiotics served as controls. Growth was visually assessed at the incubation endpoint.

## Statistical analysis

Statistical analysis for in vivo and in vitro assays was performed using GraphPad Prism 6 (version 6.07). One-way ANOVA with Tukey's multiple comparison test was applied to identify significant differences among the mean values of gene expression (Log2FC) across different shear stress conditions, and mean absorbance values across different supernatants tested on HEK-Blue IL1R cells. A two-tailed t-test was applied to identify significant differences in the mean ΔCt between bacterial inoculum and 72-h vegetations, mean biofilm, and microcolony area between 2 experimental animal groups, and mean protein band density between 2 samples. A two-tailed Mann-Whitney U test was applied to compare the median CFU, weight, and cell count values between two experimental animal groups. Two-way ANOVA was applied to identify significant differences in the mean absorbance values among of WT and *lrgAB* mutant strains grown in 2 different media. A paired t-test was applied to assess plasmid loss between CFU counts plated on BHI and BHI + spectinomycin agar from the same vegetation homogenates. Statistical data analysis of the clinical data was conducted using R (version 2024.04.2 + 764, R Core Team, Vienna, Austria). Given the uneven distribution of *fsrA* presence between the 2 cohorts, we aimed to confirm the absence of a correlation between bacteremia duration and cohort depending on gene presence. To this end, a gamma generalized linear model was applied to examine the effect of cohort within the *fsrA*-present and *fsrA*-absent groups separately. To evaluate the relationship between gene presence and various clinical and demographic variables a contingency table analysis was performed. Contingency tables were generated to summarize the association between gene presence and each of the selected variables, as well as gene presence and a cumulative score. The scoring system was established with a maximum of 7 points describing the highest disease severity. Patients received 2 points for the fulfillment of the modified Duke Criteria[74] to classify the disease as "definite IE", or 1 point when classified as "possible IE". Further to represent the severity of the individual clinical course, one point for each of the following item was received: In-hospital mortality, necessity of treatment on the ICU, as well as length of stay equaling 14 or more days. As a representation of the local extent of the infection one point for each of the following item was received: echocardiographic evidence of vegetation as well as performance of valve replacement or device extraction, such as infected pacemaker electrodes, due to IE. Fisher's exact test was used to assess the statistical significance of these associations. Heatmaps were created for each contingency table using ggplot2 to visualize the frequency of combinations between *fsrA* presence and the selected variable categories, with color gradients indicating frequency intensity. To compare the baseline characteristics of both studies, descriptive statistics were generated using the gtsummary package to summarize categorical and continuous variables by the presence or absence of *fsrA* (Supplementary Data 3). Categorical variables were summarized as counts and percentages, while continuous variables were reported as medians with interquartile ranges (IQR). Comparisons between groups were performed using the appropriate statistical tests: Chi-square tests or Fisher's exact tests for categorical variables and Mann-Whitney tests for continuous variables. Subgroup analyzes were performed for the Pittsburgh and Swiss ENVALVE cohorts, and results were compared across these cohorts. A combined summary for both cohorts was also generated. Pairwise statistical analysis of virulence gene presence/absence data was conducted by applying Fisher's exact test to all possible gene-gene combinations, calculating odds ratios and *p* values, and adjusting for multiple testing using the Benjamini-Hochberg FDR method. Significant associations (FDR < 0.05) were classified as correlations or anti-correlations, and the strength of evidence was represented as -log10 of the adjusted *p* values.

## Reporting summary

Further information on research design is available in the Nature Portfolio Reporting Summary linked to this article.

## Data availability

Source data are provided with this paper. Transcriptomic data generated in this study have been deposited in NCBI under the bioproject accession numbers PRJNA1219810 and PRJNA1219807. Illumina data for the ENVALVE cohort isolates generated in this study have been deposited in NCBI under the bioproject accession number PRJNA1333350. MS proteomic data generated in this study have been deposited to MassIVE under accession number MSV000097326. GC-MS data generated in this study have been deposited in Yareta repository under https://doi.org/10.26037/yareta:gsciqvezcffqpoxxie5qfy7nyy.This paper does not generate original codes. Source data are provided with this paper.

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

## Acknowledgements

Funding for this work was provided by the Singapore National Research Foundation and Ministry of Education Singapore under its Research Center of Excellence Program (SCELSE). Funding by Singapore Ministry of Education (MOE2019-T2-2-089) was awarded to K.A.K, and the National Medical Research Council Open Fund (MOH-000645) to K.P. and K.A.K. This study was also supported by the Wallenberg Foundation Postdoctoral Fellowship at Nanyang Technological University Singapore to H.A., SCELSE Seed Funding (SF-05) to H.A., Open Fund – Young Individual Research Grant by the National Medical Research Council Singapore (MOH-000939) to H.A., the University Of Zurich CRPP 'Personalized Medicine Of Persisting Bacterial Infections Aiming to Optimize Treatment and Outcome' to A.S.Z., B.H. and S.D.B. and by the Schweizerischer Nationalsfonds (SNF) (grant 310030_204343 to A.S.Z., grant 211422 to S.D.B, grant 32003B_219351 to B.H, and grant 310030_219227 to K.A.K.). This work was also supported by National Institutes of Health grant R21AI164018 to D.V.T., and by the Department of Medicine at the University of Pittsburgh School of Medicine. J.K. was supported through funding by a generous donor advised by CARIGEST SA (https://carigest.ch/en), acquired by Dominique Soldati-Favre. Y.L was also supported by the Pitt-Tsinghua Partnership Program. The funders had no role in study design, data collection and analysis, decision to publish, or preparation of the manuscript. We thank Gary Dunny and Jennifer Dale for providing the *E. faecalis* transposon mutants used in this study. We are grateful for insightful discussions about this project with Alex Persat (EPFL), Carey Nadell (Dartmouth), and Irina Afonina (SMART). We thank our colleagues at the SCELSE Sequencing Facility for performing library preparation and RNA sequencing. We thank the instructors René Remie and Irene Cuesta at the René Remie Surgical Skills Center for teaching us the surgical techniques required for the IE animal model. We are grateful to Antonin André (University of Geneva) for critical feedback on the manuscript.

## Author contributions

Conceptualization: H.A., K.A.K. Methodology: H.A., V.S., W.I.S., Y.L., S.D.S., J.K., S.D.B., S.L.W., D.V.T., A.S.Z., K.A.K. Formal Analysis: H.A., V.S., W.I.S., Y.L., S.D.S., J.K., S.D.B., A.Z., D.V.T., A.S.Z., K.A.K.Investigation: H.A., V.S., W.I.S., Y.L., R.J.W.T., K.K.F.N., C.J.Y.N., S.M.R., F.R.T., R.A.G.D.S., C.C.W., S.D.S., J.K., L.M.N., C.M., J.J.W. Resources: H.A., K.P., S.D.B., S.L.W., D.V.T., A.S.Z., K.A.K.Writing – Original Draft: H.A. Writing-Review and editing: V.S., W.I.S., Y.L., R.J.W.T., K.K.F.N., C.J.Y.N., S.M.R., F.R.T., R.A.G.D.S., C.C.W., S.D.S., J.K., L.M.N., C.M., J.J.W., K.P., B.H., S.D.B., S.L.W., D.V.T., A.S.Z., K.A.K.Visualization: H.A., V.S.Supervision: H.A., S.L.W., K.A.K. Funding Acquisition: H.A., K.P., B.H., S.D.B., S.L.W., D.V.T., A.S.Z, K.A.K.

## Competing interests

The authors declare no competing interests.

## Additional information

[1]Singapore Centre for Environmental Life Sciences Engineering, School of Biological Sciences, Nanyang Technological University, Singapore, Singapore. [2]Division of Infectious Diseases, University Hospital Zurich, Zurich, Switzerland. [3]Department of Medicine, University of Pittsburgh, Pittsburgh, PA, USA. [4]School of Medicine, Tsinghua University, Beijing, China. [5]Singapore-MIT Alliance for Research and Technology Centre, Singapore, Singapore. [6]Department of Microbiology and Molecular Medicine, University of Geneva, Geneva, Switzerland. [7]Lee Kong Chian School of Medicine, Nanyang Technological University, Singapore, Singapore. [8]National Center for Infectious Diseases (NCID), Singapore, Singapore. [9]Tan Tock Seng Hospital, Singapore, Singapore. ✉e-mail: Haris.Antypas@ntu.edu.sg; Kimberly.Kline@unige.ch

