## [Transparent Peer Review file · Nature Communications]

Loss of Fsr quorum sensing promotes biofilm formation and worsens outcomes in enterococcal infective endocarditis

Corresponding Author: Professor Kimberly Kline

Version 0:

Reviewer comments:

Reviewer #1

(Remarks to the Author)

The authors characterize the role of the *fsr* quorum sensing in infective endocarditis caused by *Enterococcus*. Their first major claim is that fluid flow negatively impacts the accumulation of quorum signal, which is established in other systems, and they provide convincing transcriptomic evidence for in vitro. QS is known to impact the formation of biofilms and vegetation; using a rat model, they found QS-mutant bacteria develop greater biofilm by 72 hrs, with no difference at 6 hrs. This is put together into a plausible but undersupported model by which fluid flow prevents the induction of the Fsr QS system in *E. faecalis* until biofilms become shielded from fluid flow. They go on to show bacteria mutant for gelatinase, one of the major known QS-regulated virulence factors, also develop greater biofilm, a possible advance over prior studies that have examined this in more reductionist systems to different conclusions. Gelatinase cleaves many substrates, and using an in vitro assay the authors newly describe the cleavage of the host cytokine IL-1 β . Some cleavage products of this cytokine are proinflammatory, and some lose activity, but the authors ultimately observe no difference that could be attributable to this in the infection model. This is in contrast to the title, that claims it activates inflammation. Finally, the authors perform RNAseq and across additional in vitro and in vivo assays describe *fsr* effects on metabolism, antibiotic tolerance, disease severity, and other broad effects other than gelatinase expression. Overall, I found the work to describe an advance to the enterococcal infective endocarditis field, with elements that will appeal to those with interests in quorum sensing and host pathogen interactions. However, significant work is needed to develop a “main story”. The effect of flow and biofilm on *fsr* during actual infection is not fully explored, and the effects of gelatinase and other elements regulated by *fsr* are ultimately descriptive and do not reach a mechanistic conclusion – see below.

1) The major question proposed is does flow impacts the *fsr* regulon in vivo? It is not clear why 30 min is sufficient for major QS regulation changes in vitro (Fig 1), but 6 hr is not in vivo (Fig 2) and requires 72 hr for phenotypic (Fig 3) and expression (Fig 6) differences. Is this phenotypic difference just from a cumulative effect of expression differences? This transition relating the in vitro flow condition to an in vivo phenotype where flow is not under control requires additional measures to make sense. It would make sense to examine expression in bacteria exposed to flow, eg, valve surface, vs deeper tissue or other appropriate control. Altogether, these data support the simple, established fact that QS plays a role during infection. It falls short in showing when QS is not important during infection due to flow, the part of the overall proposed mechanism that carries the most novelty. Reporters, coinfections, and inducible systems (pTCV-Ptet::gelE) may be useful for teasing out the in vivo relevance.

2) No evidence is provided on a role for IL1b in this model. Since gelatinase is only one part of the *fsr* response (Fig 1), and IL1b cleavage likely a minor part of the gelatinase response (due to its broad activity), this connection is tenuous. IL-1b cleavage is purported to promote host cell infiltration (lines 250), yet no significant cleavage is observed during infection (Fig 5B), and lack of immune cell difference in the response to vegetations (lines 225) both argue against IL-1b playing an important role. Fig 5B shows the active products observed in fig 5C do not happen during infection and not in an *fsr*-variable manner. Is this cleavage conserved between species? Additionally, no support given that the IL-1b is extracellular as claimed in line 237. Most of this is shown to be pro-IL-1b, which is expected to be intracellular, and inaccessible to gelatinase. How does gelatinase access the pro-IL1b?

3) Additionally, this IL1b cleavage event needs additional documentation. Data supporting the F104 cleavage site is absent (Line 246), so its not clear how cleavage site was identified. The gelatinase motif claimed is an identical match to additional residues in this region (FF) – why aren't they cleaved? Fig 5C observes degradation of IL1b, not a stable mature

product. Additionally, can the authors explain the loss of pro-IL1b across this experiment? This is *fsr*- and *gelE*- independent, and would appear to be the major driver of IL1b signaling potential in these conditions. The mass spec analysis of Fig 5E is also unconventional, and confusing. The model proposes a cleavage removing the N-terminus, but not clear why these peptides wouldn't be found anymore. Cleavage is expected to lead to smaller products from larger, not their complete destruction. Possibly something in the preparation is leading to this artifact where the N-terminus can no longer be observed, but at the least, controls should be included for this such as analysis with similar proteins, such as LasB or SpeB. Also, 1-51 should be examined too, to support this mechanism. Alternatively, conventional measures include N-terminal protein sequencing, or various labeling technologies, to positively identify specific sites. Protein staining, rather than immunoblotting, is recommended to detect all cleavage products formed, not just those retaining the antibody epitope.

4) Fig 7F appears under-powered, and an unexpected distribution. 7E shows ~1/2 are *fsrA*+/-, but 7F mostly focuses on isolates that are *fsr*+. Significance is reached by a sole individual with a remarkably long bacterium by an *fsr*- isolate. With a more representative sampling, this very likely is not accurate. What are the inclusion/exclusion criteria? Where any *fsr*- by means other than regional deletion? A heterogeneous population is possible - how many clones were examined for each patient? Is it *fsr*+ if even a tiny fraction of the population retains the gene, or *fsr*- if even a tiny fraction of the population mutates it?

Minor

Line 167: References 49 and 50 are redundant, and do not offer sufficient support that fibrin is a major component of vegetation. This offered mechanism that gelatinase activity on fibrin is a contributor is not obviously supported by the experiments of Fig 4, showing differences in CFU but not biomass in a gelatinase-dependent manner. This model should be better explained.

Biofilm polysaccharides should be imaged between wt, *gelE*, and *fsr* conditions since this is a known QS-regulated phenotype and known to be responsible for the purported phenotypes.

How are differences in the number of microcolonies per mouse accounted for in Fig 3E? What is the limit of detection? How does this compare to 4H where only one mouse is examined for the *gelB* mutant? How much variability between animals is expected?

The conditions for extended Fig 3C are unclear

Extended Fig 3D requires a negative control

Neutrophils are proposed to be a source of pro-IL1b for *gelE* cleavage – does this occur during enterococcus infection of neutrophils?

Reviewer #2

(Remarks to the Author)

This is an interesting manuscript by Antypas et al. that sheds new light on the role of the *Fsr* quorum sensing system in infective endocarditis caused by *Enterococcus faecalis*. The paper highlights a lack of knowledge about *Fsr* activity during fluid flow and how that contributes to vegetation development during endocarditis. The authors make the important point that quorum sensing is not only regulated by cell density, as shear/flow and spatial constraints can affect accumulation of signaling molecules. They also highlight and build upon the prior paradoxical observation from many groups that a large number of *Ef* clinical isolates have mutations in the *Far* quorum sensing system, despite the importance of this system (and the *Fsr*-regulated proteases *GelE* and *SprE*) for virulence in multiple animal models.

The authors first use RNAseq to profile transcriptional differences between *Ef* grown statically and with shear stress/flow to model blood flow. They find that *Fsr* and multiple *Fsr*-regulated genes are downregulated with fluid flow. From this, they confirm prior results showing that the *Fsr* system does not contribute to development of early vegetation development in a rat model of endocarditis, and build upon this by examining a later stage of infection. Strikingly, vegetations from animals infected with the *Fsr* mutant were significantly larger than animals infected with OG1RF. Next, they examine the role of the *Fsr*-regulated proteases *GelE* and *SprE* in modulating the host immune response during endocarditis and show that IL-1B co-localizes with *Ef* vegetations. Cleavage/generation of the active IL-1B fragment requires functional/active *GelE* (shown using multiple complementary approaches), providing insight into how *Fsr*/*GelE* may modulate the immune response during infection. The authors then investigated the impact of *Fsr* on *Ef* gene expression in late (72 hr) vegetations and found that absence of *Fsr* led to upregulation of *IrgAB* and differential expression of sugar/carbohydrate metabolic pathways, suggesting that *Fsr* quorum sensing in *Ef* is important for metabolism during endocarditis. The authors also demonstrate that the *fsr* mutant is more tolerant to gentamicin treatment, underscoring how loss of *Fsr* may provide a protective effect in vivo. Finally, the authors seek to understand why many *Ef* clinical isolates have a mutation disrupting *Fsr*, an interesting phenomenon that has been documented previously. Using existing patient records, they show a minor difference between bacteremia duration and disease severity score based on intact *fsrA* in clinical isolates.

Overall, this paper will be impactful as it aims to provide context for why a large fraction of *Ef* isolates have mutations that disrupt *Fsr* expression and activity when *Fsr* is important for virulence and disease progression. This has been an outstanding question in the field for some time, and this paper is the first to provide evidence for why these genotypes may

be detected. The methods provide an appropriate level of detail. Noteworthy results are that the loss of Fsr quorum sensing prevents control of biofilm/vegetation size and promotes tolerance to antibiotics in vivo. The in vivo RNAseq is an important data set for the field, and the fluid flow RNAseq is relevant not only for the study of endocarditis, but also other infection sites. The approaches for immunofluorescence and in vivo examination of vegetation size/structure will be relevant for other groups interested in host-pathogen dynamics during endocarditis. The authors nicely connect their results to what is known about biofilm formation and endocarditis in other Gram-positive pathogens. However, there are some sections where additional replicates, complementation, statistical analysis, and comparison to existing literature would strengthen the conclusions. My overall comments for the authors are below:

1) In some places, additional replicates and statistical comparisons would strengthen the conclusions. Statistics are missing from several panels in Extended Data Fig 2 and in Figure 4G. In Figure 4H, the microcolony areas were measured from 1 animal in 1 experiment, and it is unclear whether a direction comparison can be made to the parental strain experiments in Figure 3E. Additional animal work would not be required here if images sets from additional animals already exist.

2) The IrgAB data presented supports the conclusion that the IrgA mutant has a growth defect in M1+pyruvate, but the direct link between IrgAB and pyruvate transport would be strengthened by inclusion of pyruvate accumulation assays or measure of intracellular pyruvate. Additionally, the IrgAB experiments would be strengthened by complementation of IrgA and IrgB and additional replicates for the experiments shown in Extended Data Fig 4 (as only 1-2 replicates were done).

3) Intensity of the 35 kDa band corresponding to pro-IL-1B in Figure 5C is reduced in the gelE deletion strain relative to the fsr mutant at 4 and 6 hours. Is this due to non-specific cleavage/degradation?

4) As the authors note, as vegetations develop and interior cells become protected from/less exposed to fluid flow, this should lead to activation of quorum sensing. Presumably cells that are on the exterior of the vegetation would still experience fluid flow, less Fsr activation, and lower production of GelE. Does GelE that would be produced by these interior cells have access to substrates like IL-1B given the spatial distribution of the vegetation? Can GelE diffuse out of the core of the vegetations to the surface or can IL-1B reach the interior of the vegetation?

5) Previous microarray work identified numerous genes and metabolic pathways differentially regulated in fsrB and gelE deletion mutants depending on growth phase (PMC1446981). Is there overlap between these genes and the fsr-regulated genes identified in the RNAseq sets in this study? This could be interesting to consider given the shift in metabolism that occurs in different growth phases.

6) The link between loss of gelE/fsr and accumulation of Ace was previously published. Did the authors see any changes in Ace expression in their RNAseq? Is Ace likely to contribute to overall vegetation structure? It might be beneficial to mention briefly given that Ace contributes to endocarditis pathogenesis (PMC3165527).

7) Figure 7F is somewhat confusing as it is not clear what each data point represents. Are only 3 patients with fsrA-negative isolates shown? It would be helpful to see bacteremia duration data points for all patients (if available).

8) Based on Supp Data 3, it looks like the proportion of patients with loss of fsrA is opposite in the Zurich and Pittsburgh cohorts (fsrA not present in a majority (17/24) patients from Zurich cohort, fsrA not present in a smaller fraction (21/57) patients from Pittsburgh cohort). Can the authors speculate on why that might be the case? Are there different predominant circulating strain types in Switzerland vs USA? If so, given that bacteremia duration is similar between fsrA present/absent patients in each independent cohort, does that suggest there are other Ef genomic features driving bacteremia duration?

9) It looks like the absence of fsrA in isolates from patients (Figure 7G) results in bimodal distribution of disease severity score (1 cluster with scores 2-3, 1 cluster with scores 5-6) while presence of Fsr results in normal distribution of this metric, which is very interesting. Can the authors speculate on what might drive this?

Reviewer #3

(Remarks to the Author)

Reviewer #4

(Remarks to the Author)

The manuscript by Antypas et al. explores how the Fsr (quorum sensing (QS) system influences biofilm formation and inflammation in infective endocarditis (IE) using *Enterococcus faecalis* as model organism. QS is a bacterial communication mechanism that regulates biofilm development and other collective behaviors through autoinducer signaling molecules. QS activation, driven by increased bacterial density and various environmental factors, can either promote or inhibit gene expression related to biofilm maturation or dispersal. This study highlights that fluid flow can prevent Fsr activation, while confined environments facilitate it. The research demonstrates that the Fsr system restricts biofilm growth and triggers inflammation by upregulating gelatinase, which activates IL-1 β . Loss of the Fsr system results in unchecked biofilm growth, reduced inflammation, and increased antibiotic tolerance, correlating with severe disease outcomes.

The authors provided solid in vitro and in vivo data to support their major conclusions; the large amount of work involved in the project should also be appreciated. However, the study lacks novelty as it reiterates well-established concepts in QS and biofilm research. The role of QS in regulating biofilm formation and its impact on infection outcomes has been extensively studied in various human pathogens, including *E. faecalis*. While the specific focus on the Fsr QS system in the context of IE is valuable, the overall findings do not significantly advance our current understanding of QS mechanisms or their implications in biofilm-associated infections. General and more specific comments follow.

1. A rationale should be provided for using stationary-phase cells when incubating bacteria under fluid flow conditions.
2. Since the *fsrA* gene is under the control of a constitutive promoter, how do you explain the results obtained after exposure to fluid flow conditions? Still related to the gene expression data, it is not clear why the authors normalized Ct values obtained to the expression of *recA* gene (used as reference) instead of normalizing to the number of cells, since we can assume that at 20 dynes/cm² there are fewer cells attached.
3. While the authors concluded their first section by stating that fluid flow impacts QS and triggers physiological adaptations in *E. faecalis*, it is unclear how they formulated the hypothesis presented in the second section based on the results shown in Figure 1. The connection between the observed effects of fluid flow on QS and the proposed limitations on Fsr activation during early colonization required further clarification.
4. The section describing the impact of the absence of *gelE* and *sprE* genes on biofilm growth could be shortened. In fact, the data presented could be merged with the previous section describing the impact of the *fsr* knockout mutant on biofilm growth in late IE.
5. IE typically affects injured or abnormal heart valves, which often remain asymptomatic until a blood-borne infection occurs. The most common etiological agents are oral streptococci, *S. aureus*, and *E. faecalis*. In the case of *E. faecalis*, previous studies have found that neither the *fsr* nor gelatinase production is more common in disease-associated isolates than in isolates colonizing healthy individuals. A surface protein, *Esp*, has been described as being enriched among endocarditis and bacteremia isolates of *E. faecalis* but is rare among fecal isolates. Clinical isolates of *E. faecalis* found in urine often exhibit a chromosomal deletion involving the *fsr* locus, while some other studies have reported the presence of the *fsr* locus in endocarditis isolates. Without a deeper genomic analysis of the 81 enterococcal isolates, the results presented in Figure 7 and Extended Figure 5 add little to the manuscript.
6. Although the authors performed gene ontology enrichment analysis (Fig. 1B), it would be beneficial to provide details regarding the most important factors involved in *E. faecalis* biofilm patterns, including initial attachment, biofilm maturation, dispersal. Specifically, the prominent adhesive factor *Ebp* (endocarditis and biofilm-associated pilus) should be highlighted for its similarity to the fibronectin-binding protein found in *S. aureus*, another agent involved in IE. Other factors of interest include the cell wall anchored collagen adhesin, *EbfA*, *BgsA*, and *SagA* adhesins.

Version 1:

Reviewer comments:

Reviewer #1

(Remarks to the Author)

My comments have been addressed

Reviewer #2

(Remarks to the Author)

This is a revised manuscript by Antypas et al. investigating the role of the Fsr quorum sensing system in enterococcal infective endocarditis. The authors have thoroughly revised the manuscript and addressed major critiques from the first round of review, resulting in a compelling story. My only comment for the submitted revision is that, from my reading, whole-genome sequencing was not done on the *IrgA/B* mutants. The authors comment on problems with complementation via overexpression. However, the conclusions from this portion of the paper would be strengthened if whole-genome sequencing were done on all *IrgA/B* mutants (including the *fsr IrgAB* deletion strain) to ensure that no additional mutations are present in these strains that may confound the results. Otherwise, I think the authors have addressed all major points from the first round of reviews.

Reviewer #3

(Remarks to the Author)

We would like to thank the reviewers for their thorough evaluation of our study and for providing constructive feedback that has greatly strengthened this manuscript. In the revised manuscript, we introduced several changes, including:

- Demonstrating how fluid flow and microcolony size regulate QS activation with spatial resolution *in vivo*
- Strengthening our conclusion that loss of gelatinase promotes larger microcolonies by including additional biological replicates in our analysis
- Expanding our investigation on *lrgAB* by including GC-MS and *in vivo* data
- Providing species-specific data on pro-IL-1 β processing by gelatinase and clarifying our hypothesis on its impact on inflammation
- Deepening our genomic analysis on the clinical isolates by performing WGS on the Swiss cohort strains

These represent some of the major revisions, alongside additional clarifications and improvements throughout the manuscript, which together make the main story more explicit, add mechanistic detail, strengthen our conclusions, and temper certain claims where warranted.

Below, we provide a point-by-point response to all reviewer comments, with changes indicated in the revised manuscript in red.

REVIEWER COMMENTS

Reviewer #1 (Remarks to the Author):

*The authors characterize the role of the *fsr* quorum sensing in infective endocarditis caused by *Enterococcus*. Their first major claim is that fluid flow negatively impacts the accumulation of quorum signal, which is established in other systems, and they provide convincing transcriptomic evidence for *in vitro*. QS is known to impact the formation of biofilms and vegetation; using a rat model, they found QS-mutant bacteria develop greater biofilm by 72 hrs, with no difference at 6 hrs. This is put together into a plausible but undersupported model by which fluid flow prevents the induction of the *Fsr* QS system in *E. faecalis* until biofilms become shielded from fluid flow. They go on to show bacteria mutant for gelatinase, one of the major known QS-regulated virulence factors, also develop greater biofilm, a possible advance over prior studies that have examined this in more reductionist systems to different conclusions. Gelatinase cleaves many substrates, and using an *in vitro* assay the authors newly describe the cleavage of the host cytokine IL-1 β . Some cleavage products of this cytokine are proinflammatory, and some lose activity, but the*

authors ultimately observe no difference that could be attributable to this in the infection model. This is in contrast to the title, that claims it activates inflammation. Finally, the authors perform RNAseq and across additional *in vitro* and *in vivo* assays describe *fsr* effects on metabolism, antibiotic tolerance, disease severity, and other broad effects other than gelatinase expression. Overall, I found the work to describe an advance to the enterococcal infective endocarditis field, with elements that will appeal to those with interests in quorum sensing and host pathogen interactions. However, significant work is needed to develop a “main story”. The effect of flow and biofilm on *fsr* during actual infection is not fully explored, and the effects of gelatinase and other elements regulated by *fsr* are ultimately descriptive and do not reach a mechanistic conclusion – see below.

1) The major question proposed is does flow impacts the *fsr* regulon *in vivo*? It is not clear why 30 min is sufficient for major QS regulation changes *in vitro* (Fig 1), but 6 hr is not *in vivo* (Fig 2) and requires 72 hr for phenotypic (Fig 3) and expression (Fig 6) differences. Is this phenotypic difference just from a cumulative effect of expression differences? This transition relating the *in vitro* flow condition to an *in vivo* phenotype where flow is not under control requires additional measures to make sense. It would make sense to examine expression in bacteria exposed to flow, eg, valve surface, vs deeper tissue or other appropriate control. Altogether, these data support the simple, established fact that QS plays a role during infection. It falls short in showing when QS is not important during infection due to flow, the part of the overall proposed mechanism that carries the most novelty. Reporters, coinfections, and inducible systems (*pTCV-Ptet::gelE*) may be useful for teasing out the *in vivo* relevance.

We understand that the reviewer is asking for experimental evidence on how flow conditions regulate quorum sensing *in vivo*. We sought to address this question by demonstrating *Fsr* QS activation with spatial resolution in vegetations. We constructed an *E. faecalis* strain carrying a reporter plasmid expressing GFP under the control of the gelatinase promoter (WT *Pgele*-GFP), alongside a control strain carrying the reporter plasmid with a constitutive promoter (WT *Pcfb*-GFP). We added a new supplementary figure (**Extended Data Fig. 1A-H** in the revised manuscript) demonstrating that this reporter plasmid is specifically activated by *Fsr* QS *in vitro*. Importantly, new *in vivo* data in **Fig. 1D-F** show that *Fsr* QS activation depends on the location and size of the microcolony. Vegetation embedded microcolonies exhibit higher QS activation than superficial ones, and large microcolonies display higher activation than small ones. In contrast, WT *Pcfb*-GFP bacteria remained fluorescent regardless of location (**Extended Data Fig. 1K**). We further confirmed plasmid stability at 24 hpi *in vivo* (**Extended Data Fig. 1I-J**), ruling out plasmid loss as a cause of false-negative gelatinase expression. The first results section has been renamed to “Fluid flow and microcolony size regulate *Fsr*

QS activation in IE” and the Results section text has been revised in **lines 135-151, 154-155, 162-164, and 167-172** to reflect the addition of these new results. Relevant Methods sections have been updated to include the methodology for these new experiments (see *Bacterial cultures, DNA manipulation and construction of deletion mutants, Infective Endocarditis rat model, Tissue immunofluorescence staining, Microscopy and image analysis*). Finally, we have also added qPCR data for *fsrC* expression at 24 hpi for a more detailed analysis of its expression *in vivo* (**Fig. 3A** in the revised manuscript).

2) No evidence is provided on a role for IL1b in this model. Since gelatinase is only one part of the *fsr* response (Fig 1), and IL1b cleavage likely a minor part of the gelatinase response (due to its broad activity), this connection is tenuous. IL-1b cleavage is purported to promote host cell infiltration (lines 250), yet no significant cleavage is observed during infection (Fig 5B), and lack of immune cell difference in the response to vegetations (lines 225) both argue against IL-1b playing an important role. Fig 5B shows the active products observed in fig 5C do not happen during infection and not in an *fsr*-variable manner. Is this cleavage conserved between species? Additionally, no support given that the IL-1b is extracellular as claimed in line 237. Most of this is shown to be pro-IL-1b, which is expected to be intracellular, and inaccessible to gelatinase. How does gelatinase access the pro-IL1b?

We appreciate the reviewer’s constructive criticism of this part of the manuscript, as it has prompted us to further investigate the gelatinase effect on human and rat pro-IL-1 β . We also acknowledge that our original presentation of the results may have been confusing. Below, we first clarify the rationale in the original manuscript and then explain how we have revised the manuscript.

In the original manuscript, our aim was to suggest that gelatinase could contribute to inflammation modulation, based on our *in vitro* observation that it cleaves human pro-IL-1 β and on prior reports of other bacterial proteases (e.g., SpeB, LasB) exerting similar effects. We did not intend to imply that IL-1 β plays a central role in our *in vivo* model.

Consistent with the reviewer’s point, there is no statistically significant difference between the number of immune cells in vegetations between WT and Δ *gelE* infection (**Extended Data Fig. 2G-I** in the original manuscript, now **Extended Data Fig. 4D-F** in the revised manuscript). However, the Δ *gelE* infection leads to an increase in vegetation CFU (**Fig. 4D**), which made it puzzling to us that this was not accompanied by a corresponding increase in immune cell infiltration. Hence, in the original manuscript we proposed that Δ *gelE* biofilms might be less immunostimulatory.

Although the reviewer mentions that *no support given that the IL-1 β is extracellular*, we show that IL-1 β is localized extracellularly at the interface of biofilm and neutrophils undergoing NETosis (blue insets in **Fig. 5A** in the original and **Fig. 6A** revised manuscript).

We also provide further evidence that the neutrophils surrounding the biofilm are undergoing NETosis (**Extended Data Fig. 3A** in the original manuscript, and now **Extended Data Fig. 4A** in the revised manuscript). We therefore proposed that pro-IL-1 β becomes accessible to gelatinase through release of intracellular contents from neutrophils undergoing NETosis.

Also consistent with the reviewer's point, there is no statistically significant difference in the levels of pro-IL1 β between WT and Δfsr infection (**Fig. 5B** in the original manuscript). If cleaving or degradation of extracellular pro-IL-1 β occurs *in vivo*, it is likely restricted to the interface between biofilm and neutrophils undergoing NETosis, which would be difficult to detect by western blot. Nevertheless, this result provides additional evidence that pro-IL-1 β is present within vegetations, as shown in **Fig. 5A** of the original manuscript (**Fig. 6A** in the revised manuscript). In the revised manuscript, we have replaced the western blot comparing pro-IL-1 β levels between WT and Δfsr with a western blot comparing WT- and $\Delta gelE$ -infected vegetations (**Fig. 6B** in the revised manuscript), to align directly with **Fig. 6A**, which shows IL-1 β staining in WT- and $\Delta gelE$ -infected vegetations.

To address the reviewer's question on whether gelatinase activation of IL-1 β is conserved across species, in the revised manuscript we assayed the ability of *E. faecalis* to cleave rat pro-IL-1 β . We show that gelatinase degrades & cleaves rat pro-IL-1 β into fragments below 15 kDa (**Fig. 6C** in the revised manuscript). However, these fragments were not bioactive, as they failed to activate the IL1R reporter cell line (**Extended Data Fig. 4C** in the revised manuscript).

In summary, we show that gelatinase activation of pro-IL-1 β does not occur in rats, and we have therefore revised this section to remove any claims attributing our *in vivo* observations to this mechanism. At the same time, our *in vitro* findings with human pro-IL-1 β remain important. The ability of *E. faecalis* gelatinase to generate bioactive IL-1 β fragments in humans suggests a potential biofilm-specific mechanism of inflammation modulation that warrants further investigation. We believe this represents a novel and relevant finding for IE that should be explored in future studies, particularly in the context of host–biofilm interactions.

Please also note that:

-Results section “*E. faecalis* gelatinase cleaves and activates human IL-1 β ” is moved after result section describing *lrgAB* findings in the revised manuscript to improve the flow of the manuscript

-The text in Results section “*E. faecalis* gelatinase cleaves and activates human IL-1 β ” has been fully revised to align with the new findings on rat pro-IL-1 β degradation by gelatinase (**lines 295-338**).

-We have also revised **Fig. 5F** of the original manuscript to include a multiple alignment between the cleavage region of human, rat, and mouse IL-1 β (**Fig. 6F** in the revised manuscript).

-Discussion (**lines 406-408, 459-462**) and abstract (**lines 32-34**) have been updated to reflect that activation of pro-IL-1 β is species-specific.

-Methods sections (*IL-1 β cleaving assay, IL-1 signaling assay*) have been updated to include the new experiments.

3a) Additionally, this IL1b cleavage event needs additional documentation. Data supporting the F104 cleavage site is absent (Line 246), so its not clear how cleavage site was identified.

We appreciate this important comment as it prompted us to clarify how the cleavage site was identified. We subjected single bands of human pro-IL-1 β and cleaved fragments after incubation with the *E. faecalis* WT strain for 0, 6, and 18 h to mass spectrometry analysis (See more details in our answer to your next question). We shortlisted peptides that appeared uniquely at 6 and 18 h compared to 0h and further filtered them for abundance ($\geq 10^6$). Among these, we identified a peptide beginning at residue 105, which was one of the most abundant new peptides and consistent with the generation of a ~17 kDa fragment observed by western blotting. We therefore infer that cleavage occurs between residues 104 and 105. The cleavage site at F-F also agrees with what has been previously reported in the literature. We have included **Supplementary File 4** in the revised manuscript that contains all peptides identified at the different time points, as well as peptides unique to 6- and 18-h incubation compared to 0h. We have also revised the methods section for “IL-1 β cleaving assay” with the relevant information (**lines 847-855**).

3b) The gelatinase motif claimed is an identical match to additional residues in this region (FF) – why aren't they cleaved? Fig 5C observes degradation of IL1b, not a stable mature product. Additionally, can the authors explain the loss of pro-IL1b across this experiment? This is fsr- and gelE- independent, and would appear to be the major driver of IL1b signaling potential in these conditions. The mass spec analysis of Fig 5E is also unconventional, and confusing. The model proposes a cleavage removing the N-terminus, but not clear why these peptides wouldn't be found anymore. Cleavage is expected to lead to smaller products from larger, not their complete destruction. Possibly something in the preparation is leading to this artifact where the N-terminus can no longer be observed, but at the least, controls should be included for this such as analysis with similar proteins, such as LasB or SpeB. Also, 1-51 should be examined too, to support this mechanism. Alternatively, conventional measure include N-terminal protein sequencing, or various labeling

technologies, to positively identify specific sites. Protein staining, rather than immunoblotting, is recommended to detect all cleavage products formed, not just those retaining the antibody epitope.

We thank the reviewer for seeking additional clarity on the gelatinase-mediated cleaving of pro-IL-1 β . Before we address these comments, we would like to clarify that we subjected specific excised protein bands to mass spectrometry analysis, rather than entire gel lanes. Specifically, we analyzed the band corresponding to human pro-IL-1 β from the 0-h incubation, and the ~17 kDa bands from the 6- and 18-h incubations, as this size corresponds to the expected bioactive IL-1 β generated by canonical caspase-1 cleavage. We also provide below the Coomassie-stained gel highlighting the bands that were excised and submitted for analysis (this figure is not included in the revised manuscript).

We also want to clarify that the other 2 potential F-F cleavage sites are upstream the proposed F¹⁰⁴F¹⁰⁵ site. Below we provide the annotated protein sequence for human pro-IL-1 β (not included in the revised manuscript).

```
>sp|P01584|IL1B_HUMAN Interleukin-1 beta OS=Homo sapiens OX=9606 GN=IL1B PE=1 SV=2
```

```
MAEVP ELASEMMAY YSGNEDDL F23F24EADGPKQMKCSFQDL DLCPLDGGIQLRISDHHYSGK
```

FRQAASVVAMDKLRKMLVPCPQTFQENDLST^{F93F94}PFIFEEPI^{F104F105}DTWDNEAYVHDAPV
RSLNCTLRDSQQKSLVMSGPYELKALHLQGQDMEQQVVFMSFVQGEESNDKIPVALGLKE
KNLYLSCVLKDDKPTLQLESVDPKNYPKKKMEKRFVFNKIEINNKLFEFSAQFPNWWYIST
SQAENMPVFLGGTKGGQDITDFTMQFVSS

To address the reviewer's comment that "*the gelatinase motif claimed is an identical match to additional residues in this region (FF) – why aren't they cleaved?*", we cannot rule out that they could be cleaved. However, such cleaving would be expected to generate smaller fragments (< ~7 kDa), which are not the bioactive form of IL-1 β . Our analysis was focused on the ~17 kDa fragment corresponding to the size of mature, bioactive IL-1 β . Accordingly, we specifically excised and submitted the 17 kDa band for MS analysis, and therefore do not have data on smaller degradation products. This clarification also relates to the reviewer's additional comment regarding the absence of detectable N-terminal peptides. We interpret this not as their "complete destruction," but rather as the result of two factors: (i) small peptides generated by cleavage and potential subsequent degradation are unlikely to be retained and detected under our workflow, and (ii) our experimental design focused on excision of the ~17 kDa band, which by definition excludes low-molecular-weight products such as peptides spanning residues 1–51. We agree that complementary approaches such as N-terminal sequencing, labeling strategies, or analysis with other proteases (e.g., LasB or SpeB) would provide additional validation of the cleavage site, but these are beyond the scope of the present study. Importantly, the converging evidence from western blotting, peptide mapping, fragment size, and bioactivity retainment consistently supports F104–F105 as the most plausible site of gelatinase-mediated cleavage in human pro-IL-1 β .

The reviewer also points out that "*Fig 5C observes degradation of IL1b, not a stable mature product.*" We respectfully disagree with that observation. The ~17 kDa fragment appears by 4 h and remains detectable and stable through 24 h when pro-IL-1 β is incubated with the WT strain (**Fig. 5C** in the original manuscript, **Fig. 6E** in the revised manuscript). This stability is further supported by the Coomassie-stained gel provided above.

The reviewer is also requesting to "*explain the loss of pro-IL1b across this experiment*" and further proposes that "*This is fsr- and gelE- independent, and would appear to be the major driver of IL1b signaling potential in these conditions*". We agree that in the absence of *fsr*, *gelE*, and proteolytically inactive gelatinase, there is progressive degradation of pro-IL-1 β , which could be mediated by other unidentified factors (**Fig. 5C** in the original manuscript, **Fig. 6E** in the revised manuscript). Importantly, degradation is also observed when pro-IL-1 β is incubated in plain media, suggesting this may represent protein instability rather than a specific bacterial activity. In contrast, when functional gelatinase

is present, we observe the generation of a ~17 kDa fragment that remains stable throughout the incubation (**Fig. 5C** in original manuscript and **Fig. 6E** in the revised manuscript) and retains bioactivity (**Fig. 5D** in the original manuscript and **Fig. 6F** in the revised manuscript). Moreover, overexpression of gelatinase in the Δ *gelE* pTCV-*gelE* strain (**Extended Data Fig. 3C** in the original manuscript, **Extended Data Fig. 4G** in the revised manuscript) markedly increased cleavage of human pro-IL-1 β into the ~17 kDa fragment compared to WT, indicating that the balance between degradation and cleavage is strongly influenced by gelatinase expression levels.

We also thank the reviewer for pointing out that “*The mass spec analysis of Fig 5E is also unconventional, and confusing.*”. We have removed this figure in the revised manuscript. Instead, we provide the **Supplementary File 4** in the revised manuscript that lists all peptides detected in the 3 excised bands during the MS analysis, ranked by abundance. We also revised the Methods section for “IL-1 β cleaving assay” and the corresponding Results section (**lines 326-338**) to explain how the cleaving site was determined.

4a. Fig 7F appears under-powered, and an unexpected distribution. 7E shows ~1/2 are *fsrA*+/-, but 7F mostly focuses on isolates that are *fsr*+. Significance is reached by a sole individual with a remarkably long bacterium by an *fsr*- isolate. With a more representative sampling, this very likely is not accurate.

We would like to clarify that **Fig. 7F** includes data from all 81 patients. In the original version, we used box plots to display the median, as well as the 10th and 90th percentiles, rather than plotting each individual data point. To address the reviewer’s concern about the distribution and representation, we have revised **Fig. 7F** to display all data points alongside the median. This update allows for a clearer visualization of the data spread and shows that the observed trend is not driven by a single outlier.

4b. What are the inclusion/exclusion criteria?

We have updated the Methods section “Bacterial isolates and patient cohorts” with the inclusion and exclusion criteria (Lines 527-541).

4c. Where any *fsr*- by means other than regional deletion?

We would like to mention at this point that we have performed WGS on all the clinical isolates from the Swiss cohort for the revised manuscript, which together with the WGS data available from the US cohort enabled us to perform a more detailed genomics analysis. To address the reviewer’s question, with the exception of isolate DVT1034, which harbored a highly truncated *fsrA* gene, the remaining *fsr*-negative strains lacked both *fsrA* and *fsrB*, and exhibited absence or partial presence of *fsrC*. This pattern is characteristic of strains carrying the regional deletion that encompasses the *fsr* locus. We have included **Extended Data Fig. 6** in the revised manuscript, to illustrate the

presence of the *fsr* and other virulence factor genes across the clinical isolates used in this study. See also updated text in **lines 362-363** in Results and **lines 439-453** in Discussion.

4d. A heterogenous population is possible - how many clones were examined for each patient? Is it *fsr*+ if even a tiny fraction of the population retains the gene, or *fsr*- if even a tiny fraction of the population mutates it?

In the original manuscript, we analyzed one clone per patient, derived from a single blood sample. For the Swiss cohort, this clone derived from the last blood culture obtained from the patient. To address the possibility of within-patient heterogeneity, we have now screened additional isolates from earlier blood samples collected from the same patients during hospitalization. While this approach does not rule out subpopulations with differing *fsr* genotypes, all colonies tested from the same patient consistently showed the same *fsrA* genotype across the sampling period (**Table S3** in the revised manuscript). See updated text in **lines 381-383**. Additionally, we have added a statement to the discussion to reflect the limitation of our approach (**lines 447-453**).

Regarding classification, we classified isolates as *fsr*-positive or *fsr*-negative based on the genotype of the sequenced isolate from the US cohort or PCR-screened colony from the Swiss cohort in the original manuscript. This classification remained accurate even after we performed WGS on the Swiss cohort isolates in the revised manuscript. Given that the manuscript also investigates the role of gelatinase, we have included information on *gelE* presence for all isolates (**Extended Data Fig. 6**), as well as information on gelatinase activity for isolates from the American cohort (Methods, **lines 545-548**), with all *fsrA*-positive strains exhibiting gelatinase activity, except for DVT988 and DVT1721 strains.

Minor

1. Line 167: References 49 and 50 are redundant, and do not offer sufficient support that fibrin is a major component of vegetation. This offered mechanism that gelatinase activity on fibrin is a contributor is not obviously supported by the experiments of Fig 4, showing differences in CFU but not biomass in a gelatinase-dependent manner. This model should be better explained.

We cited two representative studies reporting the presence of fibrin in histological analyses of vegetations in the revised manuscript (**line 182** in the revised manuscript).

Regarding the role of gelatinase activity on fibrin as a potential contributor to dispersal, we would like to clarify that we do not propose this as the underlying mechanism for the observed phenotype of increased vegetation mass and CFU in our study. In fact, our data argue against such a model: we observed no significant differences in bacterial

dissemination to the blood between WT and *fsr* mutants or between WT and gelatinase mutants (**Extended Data Fig. 2B, F** in the revised manuscript). Thus, we concluded that the increased bacterial load within vegetations is unlikely to result from altered dispersal dynamics driven by gelatinase-mediated fibrin degradation.

We understand that the phrasing in that section of the manuscript may have been confusing and have revised the text. See **lines 182-186**.

2. Biofilm polysaccharides should be imaged between wt, gelE, and fsr conditions since this is a known QS-regulated phenotype and known to be responsible for the purported phenotypes.

We agree this is an interesting direction, but imaging biofilm polysaccharides *in vivo* is outside the scope of the current study and would require extensive optimization, as it is unclear which carbohydrates might be produced by *E. faecalis* in IE vegetations.

3. How are differences in the number of microcolonies per mouse accounted for in Fig 3E? What is the limit of detection? How does this compare to 4H where only one mouse is examined for the gelB mutant? How much variability between animals is expected?

We quantified the total number of microcolonies identified across three cross-sections spaced 80 μm apart per vegetation. Due to the heterogeneous distribution of bacteria within vegetations, the number of microcolonies per sample varied between 20 and 65. We have updated **Fig. 4H** to include data from 3 WT-infected and 4 *gelE*-infected vegetations. Additionally, both **Fig. 3E** and **Fig. 4H** have been updated to scatterplots to display individual microcolony sizes. To address the reviewer's question on inter-animal variability, we also include below two graphs showing the distribution of microcolony area per vegetation and the mean. These are provided for reference in this response only and are not part of the revised manuscript.

4. The conditions for extended Fig 3C are unclear

We have revised the figure legend (**Extended Data Fig. 4G** in the revised manuscript) and methods section (**lines 838-841**) to clarify the conditions.

5. Extended Fig 3D requires a negative control

Supernatants used in **Extended Data Fig. 3D** in the original manuscript (**Extended Data Fig. 4H** in the revised manuscript) were taken and diluted from the supernatants assayed in **Fig. 6F** in the revised manuscript. Except for WT supernatant, all supernatants induced no or negligible activation of the IL-1R reporter cell line. Therefore, inclusion of an additional negative control was not necessary.

6. Neutrophils are proposed to be a source of pro-IL1 β for gelE cleavage – does this occur during enterococcus infection of neutrophils?

We propose that neutrophils are a source of pro-IL-1 β based on direct and indirect evidence in the manuscript. We show that IL-1 β colocalized within neutrophils in vegetations (orange inset in **Fig. 6A** & **Extended Data Fig. 4B** in the revised manuscript). We also show that neutrophils comprise ~80% of the immune cells present within vegetations (**Extended data Fig. 4F** in the revised manuscript), which in combination with the western blot detection of pro-IL-1 β (**Fig. 6B** in the revised manuscript) further suggests that neutrophils are a source of pro-IL-1 β .

Reviewer #2 (Remarks to the Author):

This is an interesting manuscript by Antypas et al. that sheds new light on the role of the Fsr quorum sensing system in infective endocarditis caused by Enterococcus faecalis. The paper highlights a lack of knowledge about Fsr activity during fluid flow and how that contributes to vegetation development during endocarditis. The authors make the important point that quorum sensing is not only regulated by cell density, as shear/flow and spatial constraints can affect accumulation of signaling molecules. They also highlight and build upon the prior paradoxical observation from many groups that a large number of Ef clinical isolates have mutations in the Fsr quorum sensing system, despite the importance of this system (and the Fsr-regulated proteases GelE and SprE) for virulence in multiple animal models.

The authors first use RNAseq to profile transcriptional differences between Ef grown statically and with shear stress/flow to model blood flow. They find that Fsr and multiple Fsr-regulated genes are downregulated with fluid flow. From this, they confirm prior results showing that the Fsr system does not contribute to development of early vegetation development in a rat model of endocarditis, and build upon this by examining a later stage of infection. Strikingly, vegetations from animals infected with the Fsr mutant were significantly larger than animals infected with OG1RF. Next, they examine the role of the Fsr-regulated proteases GelE and SprE in modulating the host immune response during endocarditis and show that IL-1B co-localizes with Ef vegetations. Cleavage/generation of the active IL-1B fragment requires functional/active GelE (shown using multiple complementary approaches), providing insight into how Fsr/GelE may modulate the immune response during infection. The authors then investigated the impact of Fsr on Ef gene expression in late (72 hr) vegetations and found that absence of Fsr led to upregulation of lrgAB and differential expression of sugar/carbohydrate metabolic pathways, suggesting that Fsr quorum sensing in Ef is important for metabolism during endocarditis. The authors also demonstrate that the fsr mutant is more tolerant to gentamicin treatment, underscoring how loss of Fsr may provide a protective effect in vivo. Finally, the authors seek to understand why many Ef clinical isolates have a mutation disrupting Fsr, an interesting phenomenon that has been documented previously. Using existing patient records, they show a minor difference between bacteremia duration and disease severity score based on intact fsrA in clinical isolates.

Overall, this paper will be impactful as it aims to provide context for why a large fraction of Ef isolates have mutations that disrupt Fsr expression and activity when Fsr is important for virulence and disease progression. This has been an outstanding

question in the field for some time, and this paper is the first to provide evidence for why these genotypes may be detected. The methods provide an appropriate level of detail. Noteworthy results are that the loss of Fsr quorum sensing prevents control of biofilm/vegetation size and promotes tolerance to antibiotics in vivo. The in vivo RNAseq is an important data set for the field, and the fluid flow RNAseq is relevant not only for the study of endocarditis, but also other infection sites. The approaches for immunofluorescence and in vivo examination of vegetation size/structure will be relevant for other groups interested in host-pathogen dynamics during endocarditis. The authors nicely connect their results to what is known about biofilm formation and endocarditis in other Gram-positive pathogens. However, there are some sections where additional replicates, complementation, statistical analysis, and comparison to existing literature would strengthen the conclusions. My overall comments for the authors are below:

1) In some places, additional replicates and statistical comparisons would strengthen the conclusions. Statistics are missing from several panels in Extended Data Fig 2 and in Figure 4G. In Figure 4H, the microcolony areas were measured from 1 animal in 1 experiment, and it is unclear whether a direction comparison can be made to the parental strain experiments in Figure 3E. Additional animal work would not be required here if images sets from additional animals already exist.

We thank the reviewers for this comment. In the revised manuscript **Fig. 4H** now includes data from 3 WT-infected and 4 gelE-infected vegetations, with statistical analysis provided. We have also reformatted **Fig. 4H** as scatterplots to display individual microcolony sizes and statistical analysis has been applied to **Fig. 4G**. Regarding **Extended Data Fig. 2A–F** in the original manuscript, during the revision of the section on the gelatinase effect on pro-IL-1 β and the integration of new results showing that rat pro-IL-1 β is degraded rather than cleaved by gelatinase, we carefully re-evaluated these figures. In particular, we recognized that the supplementary Western blot panel was a potentially confusing and inaccurate way to estimate the ratio of immune cells to bacteria. As increased bacterial presence is already more appropriately and quantitatively addressed in **Fig. 3C** and **Fig. 4D** by CFU quantification, we omitted the panel in the revised version.

2a) The lrgAB data presented supports the conclusion that the lrgA mutant has a growth defect in M1+pyruvate, but the direct link between lrgAB and pyruvate transport would be strengthened by inclusion of pyruvate accumulation assays or measure of intracellular pyruvate.

We thank the reviewer for this suggestion. In the revised manuscript, we have added new GC-MS analyses of extracellular and intracellular pyruvate in bacteria grown in M1 supplemented with pyruvate. Under aerobic conditions, both extracellular and intracellular pyruvate were significantly higher in the *lrgA::Tn* strain compared to WT and *lrgB::Tn* (**Fig. 5E** in the revised manuscript), indicating that *lrgA* plays a role in pyruvate import and downstream catabolism. We further quantified extracellular and intracellular lactate and alanine and observed no significant differences under either aerobic or anaerobic conditions (**Extended Data Fig. 3A, B** in the revised manuscript), showing that pyruvate conversion into these metabolites remained intact. Together, these new data strengthen the conclusion that *lrgAB* functions in pyruvate catabolism rather than transport.

To further explore the role of *lrgAB* in IE pathogenesis, we infected rats with a $\Delta fsr \Delta lrgAB$ double mutant. Vegetation weight and CFU were comparable to WT (**Fig. 5G, H** in the revised manuscript), indicating that *lrgAB* contributes to the increased vegetation mass and bacterial burden seen in the absence of QS. No differences were observed in spleen, liver, or blood CFU (**Extended Data Fig. 3F–H** in the revised manuscript).

The Results section presenting the *lrgAB* findings has been renamed to “*fsr* locus absence enhances metabolic adaptation via *lrgAB* and pyruvate catabolism” and the Results text has been revised to reflect the addition of new results (**lines 260-277, 282-283, 285-293**). Discussion has been revised (**lines 473-481**) and a new Methods section “Relative metabolite quantification by gas chromatography-mass spectrometry (GC-MS)” has been added.

2b) Additionally, the *lrgAB* experiments would be strengthened by complementation of *lrgA* and *lrgB* and additional replicates for the experiments shown in Extended Data Fig 4 (as only 1-2 replicates were done).

We attempted to complement the *lrgA::Tn* and *lrgB::Tn* mutants, but overexpression of *lrgAB* on a plasmid caused a pronounced growth defect, particularly in *lrgB::Tn*, even in rich media. This effect may reflect the combined impact of kanamycin selection and *lrgAB* overexpression. Although this is an interesting finding that could provide further insight on *lrgAB* function, it requires additional optimization beyond the scope and timeline of the current revision.

Regarding the Triton X-100 sensitivity assay (**Extended Data Fig. 4A** in the original manuscript, **Extended Data Fig. 3C** in the revised manuscript), we have now performed 2 additional replicates, which we include below for the reviewer’s reference. The results are in line with those presented in the manuscript, showing no lysis across strains with or without Triton X-100. We have revised the figure legend to clarify that a representative result from N = 3 is shown.

3) Intensity of the 35 kDa band corresponding to pro-IL-1B in Figure 5C is reduced in the *gelE* deletion strain relative to the *fsr* mutant at 4 and 6 hours. Is this due to non-specific cleavage/degradation?

We thank the reviewer for this observation. Our explanation to reviewer #1, question 3b, is highly relevant here:

“We agree that in the absence of *fsr*, *gelE*, and proteolytically inactive gelatinase, there is progressive degradation of pro-IL-1 β , which could be mediated by other unidentified factors (**Fig. 5C** in the original manuscript, **Fig. 6E** in the revised manuscript). Importantly, degradation is also observed when pro-IL-1 β is incubated in plain media, suggesting this may represent protein instability rather than a specific bacterial activity. In contrast, when functional gelatinase is present, we observe the generation of a ~17 kDa fragment that remains stable throughout the incubation (**Fig. 5C** in original manuscript and **Fig. 6E** in the revised manuscript) and retains bioactivity (**Fig. 5D** in the original manuscript and **Fig. 6F** in the revised manuscript). Moreover, overexpression of gelatinase in the Δ *gelE* pTCV-*gelE* strain (**Extended Data Fig. 3C** in the original manuscript, **Extended Data Fig. 4G** in the revised manuscript) markedly increased cleavage of human pro-IL-1 β into the ~17 kDa fragment compared to WT, indicating that the balance between degradation and cleavage is strongly influenced by gelatinase expression levels.”

In addition, we cannot exclude the possibility that loss of gelatinase induces compensatory expression of another protease.

4) As the authors note, as vegetations develop and interior cells become protected from/less exposed to fluid flow, this should lead to activation of quorum sensing.

Presumably cells that are on the exterior of the vegetation would still experience fluid flow, less Fsr activation, and lower production of GelE. Does GelE that would be produced by these interior cells have access to substrates like IL-1B given the spatial distribution of the vegetation? Can GelE diffuse out of the core of the vegetations to the surface or can IL-1B reach the interior of the vegetation?

Thank you for this comment. Before we address it, we would like to add that our hypothesis that microcolony localization in vegetation and exposure to flow will affect QS activity is now supported experimentally in **Fig. 1D-F** in the revised manuscript. Specifically, we demonstrate that QS expression is higher in microcolonies embedded within the vegetation compared to superficial ones. Moreover, QS activity is also dependent on microcolony size, with larger colonies showing high levels of QS expression regardless of their location.

To address the reviewer's point, **Fig. 6A** shows that neutrophils infiltrate vegetations and surround biofilms embedded within them. We further show that IL-1 β is present extracellularly at the interface between biofilms and neutrophil-derived NETs (**Fig. 6A**). Thus, while diffusion of gelatinase outward or IL-1 β inward may occur, our data support a model in which pro-IL-1 β activation is largely localized to sites of direct interaction between host immune cells and biofilm communities. Although in the revised manuscript we show that in the rat model gelatinase degrades pro-IL-1 β without generating a bioactive fragment (**Fig. 6C** and **Extended Data Fig. 4C** in the revised manuscript), we expect that the comparable distribution of neutrophils around embedded biofilms in human vegetations would provide the spatial conditions for GelE to encounter and cleave human pro-IL-1 β .

5) Previous microarray work identified numerous genes and metabolic pathways differentially regulated in fsrB and gelE deletion mutants depending on growth phase (PMC1446981). Is there overlap between these genes and the fsr-regulated genes identified in the RNAseq sets in this study? This could be interesting to consider given the shift in metabolism that occurs in different growth phases.

Thank you for this comment. We focused our comparison on the set of genes previously reported as negatively regulated by fsr (Bourgogne et al, 2008, PMC1446981), since our study focuses on the impact of the absence of the fsr locus. Out of the 97 negatively regulated genes identified in Bourgogne et al, 2008, 49 overlapped with our RNAseq dataset comparing the transcriptomes of vegetations infected with WT or Δ fsr strains at 72 hpi. These overlapping genes did not exhibit strong growth phase-dependent differences in Bourgogne et al, 2008. They were consistently negatively regulated regardless of the growth phase.

Among these, we highlight below the genes from our dataset showing >2-fold change in the absence of *fsr* in vegetations and are negatively regulated by *fsr* in Bourgogne et al., 2008:

ID	old_locus_tag	gene_name	Description	Bourgogne et al., 2008			Antypas et al., 2025		
				Fold (Late log)	Fold (Entry Stat)	Fold (Early Stat)	logFC	Pvalue	FDR
EF0292	OG1RF_10235		PTS system component	-3	-4		4.55023	0.00029	0.007
EF0105	OG1RF_10100	argF	ornithine carbamoyltransferase, catabolic	-6	-2		2.96161	0.00088	0.01
EF3327	OG1RF_12572		citrate transporter	-4			2.784	0.00092	0.01
EF1662	OG1RF_11374	buk	butyrate kinase	-11			2.69472	0.00792	0.031
EF0104	OG1RF_10099	arcA	arginine deiminase	-5	-3		2.5987	0.00439	0.022
EF0106	OG1RF_10101	arcC	carbamate kinase	-3	-3	-4	2.40184	0.00094	0.01
EF0108	OG1RF_10103		conserved hypothetical protein	-3	-3	-3	2.15922	9E-05	0.004
EF1803	OG1RF_11512		PTS system component	-2			2.07382	0.0122	0.039
EF1206	OG1RF_10978		malate oxidoreductase	-2	-5		2.00827	0.00178	0.015

arcA, *arcC*, *argF* genes belong to the arginine deiminase (ADI) pathway which allows bacteria to degrade L-arginine under anaerobic conditions to generate ATP, while others encode sugar transporters (PTS components). Similarly, butyrate kinase is involved in ATP generation under anaerobic conditions. These findings suggest that *fsr* loss derepresses multiple metabolic functions that could enhance adaptation to nutrient and oxygen availability during infection. We also note that *lrgAB* was not identified as differentially expressed in the Bourgogne et al. study, despite being upregulated in the absence of *fsr* in our *in vivo* dataset. This suggests that some aspects of *fsr*-dependent regulation may be specific to the host infection environment and not apparent under *in vitro* growth conditions.

We have revised our Discussion to include the comparison with the findings from Bourgogne et al., 2008. See **lines 486-493**.

6) The link between loss of *gelE/fsr* and accumulation of *Ace* was previously published. Did the authors see any changes in *Ace* expression in their RNAseq? Is *Ace* likely to contribute to overall vegetation structure? It might be beneficial to mention briefly given that *Ace* contributes to endocarditis pathogenesis

(PMC3165527).

We thank the reviewer for raising this interesting point and for referring us to the relevant publication. Pinkston et al. (PMC3165527) demonstrated that gelatinase cleaves Ace from the cell surface. Regarding transcriptional regulation, they reported that “*preliminary analysis of ace transcript abundance at different growth phases comparing the Δ fsrB strain to the wild-type strain showed no obvious differences*” and that “*a post-transcriptional mechanism might be responsible for the Ace surface display phenotype seen in the fsr mutants.*” This is consistent with prior microarray analyses that did not identify ace as transcriptionally regulated by Fsr *in vitro* (Bourgogne et al., 2008).

In our study, we observed changes in ace transcript levels under different conditions. Specifically, RNAseq data revealed that ace expression was downregulated in the Δ fsr strain compared to WT in vegetations at 72 hpi (LogFC = -1.8, **Supplementary File 2**). Taken together with the findings of Pinkston et al. and Bourgogne et al., our results suggest that the impact of fsr on ace expression differs between *in vitro* and *in vivo* contexts, potentially reflecting environment-specific regulatory mechanisms.

We also found that ace presence was significantly correlated with fsr presence ($p = 2e-04$) across the clinical isolates in our study, with only 2 fsr-negative strains harbouring ace (**Extended Data Fig. 6**). Thus, although ace is an important virulence factor in IE, it is unlikely to account for the increased biofilm formation, prolonged bacteremia, and higher disease severity observed in our study, as it was virtually absent in strains lacking fsrA.

7) Figure 7F is somewhat confusing as it is not clear what each data point represents. Are only 3 patients with fsrA-negative isolates shown? It would be helpful to see bacteremia duration data points for all patients (if available).

We have revised **Fig. 7F** to display all data points alongside the median. Kindly see response to question 4a from reviewer #1.

8a. Based on Supp Data 3, it looks like the proportion of patients with loss of fsrA is opposite in the Zurich and Pittsburgh cohorts (fsrA not present in a majority (17/24) patients from Zurich cohort, fsrA not present in a smaller fraction (21/57) patients from Pittsburgh cohort). Can the authors speculate on why that might be the case? Are there different predominant circulating strain types in Switzerland vs USA?

We appreciate the reviewer’s observation. In the Pittsburgh cohort, the predominant sequence types were ST6 (fsr⁺), ST179 (fsr⁻), and ST40 (fsr⁺), which helps explain the overall lower frequency of fsrA loss. The Zurich cohort is smaller and thus limits our ability to confidently determine dominant strain types. Nevertheless, we note that ST6 (fsr⁺),

which was predominant in Pittsburgh, is absent from the Zurich isolates. This absence may contribute to the higher proportion of *fsrA*-negative strains observed in the Zurich cohort. We have updated the Results sections (**lines 362-363**) and added new **Extended Data Fig. 5** to illustrate the different sequence types.

8b. If so, given that bacteremia duration is similar between *fsrA* present/absent patients in each independent cohort, does that suggest there are other *Ef* genomic features driving bacteremia duration?

We thank the reviewer for this comment. The main reason that the bacteremia duration is similar between *fsrA*-positive and *fsrA*-negative infections within each cohort is likely due to limited sample sizes, particularly in the Zurich cohort. To further address this question, we experimented expanding the Pittsburgh cohort from 57 to 119 isolates by incorporating enterococcal IE genome sequences recently published by two of our co-authors (<https://doi.org/10.1093/infdis/jiaf272>), while our current manuscript was under review. Within the expanded Pittsburgh cohort, bacteraemia duration was significantly longer in patients infected with *fsrA*-negative strains compared to *fsrA*-positive strains ($p = 0.02$). However, combining this expanded dataset with the Zurich cohort introduced cohort bias, and we therefore chose not to include this analysis in the revised manuscript. However, we include the plot comparing bacteremia duration within the Pittsburgh cohort below for the reviewer's reference.

Moreover, to determine whether other *E. faecalis* genomic features might underlie the observed differences in clinical outcomes, we examined virulence factors that anticorrelate with *fsrA* presence. We found significant anticorrelations between *fsrA* and *esp*, *EF_0149*, and some cytolysin operon genes (**Extended Data Fig. 8** in the revised manuscript). However, none of these genes were significantly associated with prolonged

bacteraemia or increased disease severity in our cohort (**Extended Data Fig. 9**). These findings support the conclusion that the absence of the *fsr* locus itself, rather than co-occurring virulence factors, is the most consistent genomic feature associated with more severe clinical outcomes and prolonged bacteremia. See also updated results text (**lines 381-390**).

9) It looks like the absence of *fsrA* in isolates from patients (Figure 7G) results in bimodal distribution of disease severity score (1 cluster with scores 2-3, 1 cluster with scores 5-6) while presence of *Fsr* results in normal distribution of this metric, which is very interesting. Can the authors speculate on what might drive this?

We thank the reviewer for this observation. While there is a bimodal distribution of disease severity among patients infected with *fsrA*-negative isolates, we currently cannot determine the underlying cause. When we examined only the *fsrA*-negative isolates ordered by disease severity, we found no consistent patterns in the presence or absence of known virulence genes or sequence type. This suggests that the observed heterogeneity in clinical outcomes among *fsrA*-negative infections may be driven by factors beyond the set of genes we analyzed or patient variables that remain to be elucidated.

Reviewer #3 (Remarks to the Author):

Reviewer #4 (Remarks to the Author):

*The manuscript by Antypas et al. explores how the *Fsr* (quorum sensing (QS) system influences biofilm formation and inflammation in infective endocarditis (IE) using *Enterococcus faecalis* as model organism. QS is a bacterial communication mechanism that regulates biofilm development and other collective behaviors through autoinducer signaling molecules. QS activation, driven by increased bacterial density and various environmental factors, can either promote or inhibit gene expression related to biofilm maturation or dispersal. This study highlights that fluid flow can prevent *Fsr* activation, while confined environments facilitate it. The research demonstrates that the *Fsr* system restricts biofilm growth and triggers inflammation by upregulating gelatinase, which activates IL-1 β . Loss of the *Fsr* system results in unchecked biofilm growth, reduced inflammation, and increased antibiotic tolerance, correlating with severe disease outcomes.*

The authors provided solid in vitro and in vivo data to support their major conclusions; the large amount of work involved in the project should also be appreciated. However, the study lacks novelty as it reiterates well-established concepts in QS and biofilm research. The role of QS in regulating biofilm formation and its impact on infection outcomes has been extensively studied in various human pathogens, including E. faecalis. While the specific focus on the Fsr QS system in the context of IE is valuable, the overall findings do not significantly advance our current understanding of QS mechanisms or their implications in biofilm-associated infections. General and more specific comments follow.

We thank reviewer #4 for reviewing our manuscript but we respectfully disagree with the overall assessment that “*the study lacks novelty as it reiterates well-established concepts in QS and biofilm research*” and that “*while the specific focus on the Fsr QS system in the context of IE is valuable, the overall findings do not significantly advance our current understanding of QS mechanisms or their implications in biofilm-associated infections*”.

As we highlighted in our introduction, while the virulence factors that facilitate initial adhesion to thrombus during early enterococcal IE have been identified, the regulation of biofilm formation and its impact on disease progression remain largely overlooked. Moreover, previous *in vitro* studies established the Fsr QS system as a positive regulator of biofilm formation. Our work directly contradicts this long-standing concept by demonstrating that loss of Fsr promotes biofilm growth and worsens infection outcome. Our study highlights that the role of QS in *E. faecalis* is context-dependent, and to our knowledge, this is the first study to define the impact of *fsr* absence in enterococcal IE. Given the homology between Fsr in *E. faecalis* and the Agr system in *S. aureus*, our findings also have broader implications for understanding the role of QS in staphylococcal IE, where similar context-dependent regulatory effects may be at play. Finally, despite extensive literature proposing anti-QS strategies as potential therapies against biofilm-associated infections, our results demonstrate that QS does not universally promote biofilm formation. In the case of enterococcal IE, Fsr activity actually restricts biofilm growth. This underscores the importance of context when considering QS as a therapeutic target and cautions against assuming that QS inhibition will always be beneficial in treating biofilm-related infections.

Please see our answers to the specific comments below.

1. A rationale should be provided for using stationary-phase cells when incubating bacteria under fluid flow conditions.

We thank the reviewer for this comment. We chose to use stationary-phase *E. faecalis* cells for the flow chamber experiments because vegetations represent a nutrient-limited environment that more closely resembles stationary-phase physiology rather than exponential growth, which primarily reflects an *in vitro* state in nutrient-rich media.

2. Since the *fsrA* gene is under the control of a constitutive promoter, how do you explain the results obtained after exposure to fluid flow conditions? Still related to the gene expression data, it is not clear why the authors normalized Ct values obtained to the expression of *recA* gene (used as reference) instead of normalizing to the number of cells, since we can assume that at 20 dynes/cm² there are fewer cells attached.

fsrA is under a constitutive promoter that sustains basal expression, but further upregulation of the Fsr QS system requires accumulation and sensing of the GBAP autoinducer. Fluid flow reduces GBAP accumulation through advection, thereby preventing upregulation of the *fsr* regulon, explaining our results in **Fig. 1**.

Regarding the question about qPCR, we understand that the reviewer refers to **Fig. 1C**. To clarify, we used the geometric mean of *recA* and *dnaB*, based on their stable expression under both static and fluid flow conditions as indicated by transcriptomic data and validation using the Bestkeeper v1 analysis tool. This information is present in the original and revised manuscript in Methods section “Quantitative PCR”.

Second, we used equal amounts of total RNA from static and shear-stress samples for cDNA synthesis (we have emphasized that in the qPCR result section in the revised manuscript). Moreover, normalization to stably expressed reference genes such as *recA* and *dnaB* is required to compare gene expression across different conditions, because it corrects for differences in RNA yield, cDNA synthesis efficiency, and template input. In contrast, normalizing to cell numbers would not account for these variables and would therefore not provide valid expression comparisons.

3. While the authors concluded their first section by stating that fluid flow impacts QS and triggers physiological adaptations in *E. faecalis*, it is unclear how they formulated the hypothesis presented in the second section based on the results shown in Figure 1. The connection between the observed effects of fluid flow on QS and the proposed limitations on Fsr activation during early colonization required further clarification.

We thank the reviewer for this comment. In the revised manuscript, we have added experimental evidence on how the fluid flow affects QS expression *in vivo* to further support the hypothesis formulated in the second results section.

As we also replied to question #1 from reviewer #1, “We constructed an *E. faecalis* strain carrying a reporter plasmid expressing GFP under the control of the gelatinase promoter (WT P_{gele}-GFP), alongside a control strain carrying the reporter plasmid with a constitutive promoter (WT P_{cfb}-GFP). We added a new supplementary figure (**Fig. S1A-H**) demonstrating that this reporter plasmid is specifically activated by Fsr QS *in vitro*. Importantly, new *in vivo* data in **Fig. 1D-F** show that Fsr QS activation depends on the location and size of the microcolony. Vegetation embedded microcolonies exhibit higher QS activation than superficial ones, and large microcolonies display higher activation than small ones. In contrast, WT P_{cfb}-GFP bacteria remained fluorescent regardless of location (**Fig. S1K**). We further confirmed plasmid stability at 24 hpi *in vivo* (**Fig. S1I-J**), ruling out plasmid loss as a cause of false-negative gelatinase expression.”

We have also revised the text in the Results sections (**lines 135-151, 154-155, 162-164, 167-171**) to provide a clearer transition from the first to the second and third section.

Overall, these new data provide support for our hypothesis that the combination of fluid flow and low population density during early colonization restricts GBAP accumulation superficial microcolonies, thereby limiting Fsr activation in superficially adhered bacteria.

4. The section describing the impact of the absence of gelE and sprE genes on biofilm growth could be shortened. In fact, the data presented could be merged with the previous section describing the impact of the fsr knockout mutant on biofilm growth in late IE.

We thank the reviewer for the suggestion but we prefer to keep the sections on downstream genes modulated by the *fsr* locus (e.g., *gelE/sprE*, *lrgAB*) separate, in order to emphasize their individual contributions to IE in the presence or absence of *fsr*.

5a. IE typically affects injured or abnormal heart valves, which often remain asymptomatic until a blood-borne infection occurs. The most common etiological agents are oral streptococci, S. aureus, and E. faecalis. In the case of E. faecalis, previous studies have found that neither the fsr nor gelatinase production is more common in disease-associated isolates than in isolates colonizing healthy individuals. A surface protein, Esp, has been described as being enriched among endocarditis and bacteremia isolates of E. faecalis but is rare among fecal isolates.

As noted in the Discussion section of the manuscript, we do state that the *fsr* locus is variably present among *E. faecalis* strains isolated from infective endocarditis, blood, urine, and feces, and we cite the study the reviewer refers to (**lines 433-435**). However, we also emphasize that the role of *fsr* in influencing disease severity has not been

previously explored (lines 433-435). Our study explicitly addresses this gap by demonstrating that the absence of the *fsr* locus correlates with larger biofilms and greater antibiotic tolerance *in vivo* and is also associated with prolonged bacteraemia and higher disease severity scores in patients with infective endocarditis.

Regarding the reviewer's comment about *esp*, we did not observe enrichment of this gene in our IE cohorts, as its frequency was approximately 50% (**Extended data Fig. 6** in the revised manuscript). In our analysis, *esp* was significantly anticorrelated with *fsr* ($p = 4.6 \times 10^{-8}$, **Extended Data Fig. 8** in the revised manuscript), but it was not associated with prolonged bacteremia or high disease severity scores (**Extended Data Fig. 8** in the revised manuscript). Moreover, the OG1RF strain used in our *in vitro* and *in vivo* experiments does not harbor *esp*, further supporting that *esp* is not implicated in the phenotypes we observed.

5b. Clinical isolates of E. faecalis found in urine often exhibit a chromosomal deletion involving the fsr locus, while some other studies have reported the presence of the fsr locus in endocarditis isolates. Without a deeper genomic analysis of the 81 enterococcal isolates, the results presented in Figure 7 and Extended Figure 5 add little to the manuscript.

As part of the revision, we have sequenced all isolates from the Zurich cohort. Combined with the previously sequenced isolates from the Pittsburgh cohort, we now provide a comprehensive genomic analysis of all 81 enterococcal strains. Specifically, we report the presence of additional virulence factors (**Extended Data Fig. 6** in the revised manuscript), show that *esp* and EF0149 are significantly anticorrelated with *fsr* but do not account for the observed phenotypes (**Extended Data Fig. 9** in the revised manuscript), and present the sequence types for all isolates (**Extended Data Fig. 5** in the revised manuscript). Collectively, these data strengthen our conclusion that the absence of *fsr* is associated with prolonged bacteremia and higher disease severity scores.

6. Although the authors performed gene ontology enrichment analysis (Fig. 1B), it would be beneficial to provide details regarding the most important factors involved in E. faecalis biofilm patterns, including initial attachment, biofilm maturation, dispersal. Specifically, the prominent adhesive factor Ebp (endocarditis and biofilm-associated pilus) should be highlighted for its similarity to the fibronectin-binding protein found in S. aureus, another agent involved in IE. Other factors of interest include the cell wall anchored collagen adhesin, EbfA, BgsA, and SagA adhesins.

We thank the reviewer for this comment. In our dataset comparing gene expression between static and fluid flow conditions, these adhesion factors did not show striking changes: *ace* was modestly upregulated (1.73-fold), while the *ebp* pilus operon was slightly downregulated (*ebpA* -0.54, *ebpB* -0.57, *ebpC* -0.72). *ebfA* (OG1RF_11019), *bgsA*

(OG1RF_12193), and sagA-like genes (salA, OG1RF_12331; salB, OG1RF_10281) were not differentially expressed. This data is already provided in **Supplementary File 1** in the original manuscript for readers to explore in detail.

Given that no major differential expression was observed for these adhesins and that the primary focus of this work is quorum sensing-mediated regulation, we have not expanded this section further. We agree that adhesion under flow is an important aspect of *E. faecalis* biology and will address it in detail in a future manuscript currently in preparation.